# Fault valving and pore pressure evolution in simulations of earthquake sequences and aseismic slip

Weiqiang Zhu [1✉], Kali L. Allison [1,2], Eric M. Dunham [1,3] & Yuyun Yang[3]

Fault-zone fluids control effective normal stress and fault strength. While most earthquake models assume a fixed pore fluid pressure distribution, geologists have documented fault valving behavior, that is, cyclic changes in pressure and unsteady fluid migration along faults. Here we quantify fault valving through 2-D antiplane shear simulations of earthquake sequences on a strike-slip fault with rate-and-state friction, upward Darcy flow along a permeable fault zone, and permeability evolution. Fluid overpressure develops during the interseismic period, when healing/sealing reduces fault permeability, and is released after earthquakes enhance permeability. Coupling between fluid flow, permeability and pressure evolution, and slip produces fluid-driven aseismic slip near the base of the seismogenic zone and earthquake swarms within the seismogenic zone, as ascending fluids pressurize and weaken the fault. This model might explain observations of late interseismic fault unlocking, slow slip and creep transients, swarm seismicity, and rapid pressure/stress transmission in induced seismicity sequences.

[1] Department of Geophysics, Stanford University, Stanford, CA 94305, USA. [2] Department of Geology, University of Maryland, College Park, MD 20742, USA. [3] Institute for Computational and Mathematical Engineering, Stanford University, Stanford, CA 94305, USA. ✉email: zhuwq@stanford.edu

Fault shear strength $\tau = f \times (\sigma - p)$ is controlled by both friction coefficient $f$ and effective normal stress $\sigma - p$, the difference between compressive total normal stress $\sigma$ and pore pressure $p$. Much attention in the earthquake modeling community has been placed on friction over the past decades, with specific focus on rate- and state-dependent effects that control the stability of sliding, as well as additional dynamic weakening processes that are likely relevant at coseismic slip velocities. With some exceptions[1–9], less attention has been placed on pore pressure dynamics, and most earthquake simulations use pore pressure (or really effective stress) as a tuning parameter chosen to produce reasonable stress drops and slip per event[10,11].

Continental strike-slip faults like the San Andreas (CA) and Alpine (New Zealand) faults can act as conduits or at least guides for mantle-derived fluid, fluids released during metamorphic dehydration reactions, and meteoric fluid that circulates in the upper crust[12–16]. The fluid transport properties of fault zones are highly variable, as a consequence of differences in structure, lithology and composition, stress state, and deformation history[17,18]. For most mature faults in crystalline rocks, fault permeability is anisotropic and varies with distance normal to the fault core, with the low permeability core acting as a barrier to across-fault flow and the high permeability damage zone facilitating upward flow along the fault[13,19,20]. Pressure gradients that exceed the hydrostatic gradient induce flow along faults, and fluid overpressure is one of the classic explanations for the weakness of the San Andreas and other plate boundary faults[13,21]. Fluids are even more important in subduction zones, owing to dehydration reactions at depth as well as overpressure from burial of sediments in the uppermost portion of the seismogenic zone[22]. Fluids and pore pressure influence fault strength and can trigger seismicity, as evidenced in both energy production activities[23] as well as naturally occurring swarm seismicity[24–28]—which might also involve fluid-driven aseismic slip[29].

Fluid flow and pore pressure are likely to be dynamic quantities, particularly near the base of the seismogenic zone, over earthquake cycle time scales. Geologists document mineral-filled veins that provide evidence for episodic fluid pressurization events in which pore pressure locally exceeds the least principal compressive total stress[26,30,31]. The intermittency of fluid pressurization and release, a concept known as fault valving[30,31], is a consequence of feedback between fault slip and deformation, which typically elevate permeability[32–34], and healing and sealing processes, like pressure solution transfer, that reduce permeability[35–39]. These feedback effects are amplified by nonlinear dependence of permeability on effective normal stress due to mechanical compression of pores and microfractures[13,19,40].

In this work, we test the viability of fault valving and explore phenomena that arise from the coupling between fluids and faulting processes. This is done in an earthquake sequence model that accounts for the coupling between fault zone fluid flow, pore pressure and permeability evolution, and elastic stress transfer from fault slip. In certain parts of parameter space, this model produces fault valving cycles of overpressure build-up and release. The model also predicts the spontaneous generation of fluid-driven aseismic slip and overpressure pulses near the base of the seismogenic zone and earthquake swarms within the seismogenic zone, as ascending fluids pressurize and weaken the fault. These phenomena might explain many related observations, such as late interseismic fault unlocking, slow slip, swarm seismicity, and rapid pressure transmission along faults in induced seismicity sequences.

## Results

**Model.** The purpose of this study is to introduce a quantitative simulation framework in which to explore the two-way coupling between fluid transport, pore pressure evolution, and fault slip over the earthquake cycle. Our focus is on the processes and phenomena that arise from this coupling, in a generic sense. We do this in the context of a quasi-dynamic[41] two-dimensional antiplane shear model of a vertical strike-slip fault in a uniform elastic half-space (Fig. 1a), the classic idealization for investigation of processes controlling earthquake sequences and aseismic slip. While parameter choices are chosen to be reasonably representative of continental strike-slip plate boundary settings, we are not attempting to model any specific fault or earthquake sequence. Furthermore, it is possible that key findings might be relevant to other tectonic settings like subduction zones. The fault obeys rate-and-state friction with a transition from velocity weakening (VW) to velocity strengthening at about 17 km depth (see also "Methods" and Table 1). The solid is loaded at a constant plate rate $V_p$ by displacement of the remote side boundaries.

Fluids migrate vertically along a tabular, porous fault zone, as in Rice's model[21], with the surrounding country rock assumed impermeable. Many studies have established that damage zone permeability is vastly higher than the surrounding country rock[13,19,20,42]. Conservation of fluid mass, Darcy's law, and linearized descriptions of fluid and pore compressibility (with an elastic matrix) give rise to a one-dimensional (1D) diffusion equation for pore pressure:

$$n\beta \frac{\partial p}{\partial t} = \frac{\partial}{\partial z}\left[\frac{k}{\eta}\left(\frac{\partial p}{\partial z} - \rho g\right)\right], \qquad (1)$$

where $n$ is the pore volume fraction, $\beta$ is the sum of fluid and pore compressibility[1,3], $k$ is permeability, $\eta$ is fluid viscosity, $\rho$ is fluid density, $g$ is gravity, and $z$ is the distance from the Earth's surface (positive down). Our 1D fluid transport model approximately captures pressure evolution and flow over time scales that are longer than the hydraulic diffusion time across the damage zone, which we estimate to be of order of days to weeks for representative damage zone properties. This 1D treatment neglects pressure gradients in the fault normal direction that arise over coseismic time scales from thermal pressurization[3] and poroelastic effects[43–45] from localized shearing or slip within the fault core. Inelastic changes in pore volume fraction (and storage $n\beta$), which can arise from shear-induced dilatancy[1], mineral precipitation in pores and microfractures[35–37], and viscous flow of the matrix[46,47], have been neglected for simplicity, and because we anticipate that changes in permeability will be more significant. The vertical (positive upward) fluid flux (volume of fluid per unit horizontal cross-sectional area per unit time) is

$$q = \frac{k}{\eta}\left(\frac{\partial p}{\partial z} - \rho g\right). \qquad (2)$$

Equation (1) requires two boundary conditions, which we take as $p = 0$ at $z = 0$ (atmospheric pressure at Earth's surface, set to zero) and $q = q_0$ (constant) at the bottom of the simulation domain placed well below the seismogenic zone. The latter is a crude approximation for a fluid source at depth and avoids more sophisticated descriptions of fluid-producing dehydration reactions, meteoric water input from the crust surrounding the fault, and other sources. We set $q_0 = 3 \times 10^{-9} \, \mathrm{m \, s^{-1}}$, which is within the range of fluxes inferred for continental plate boundary faults[12,16].

Note that $q = 0$ for the hydrostatic condition $p = \rho g z$, whereas fluid overpressure leads to upward flow ($q > 0$): $p = (\rho g + \eta q/k)z$ for constant $q$ and $k$. However, permeability $k$ is unlikely to be constant. Many experiments show that permeability decreases as effective normal stress increases, due to mechanical closure of fractures and pores[13,19,40,42]. We capture this effect as

$$k = k_{\min} + (k^* - k_{\min})e^{-(\sigma - p)/\sigma^*}, \qquad (3)$$

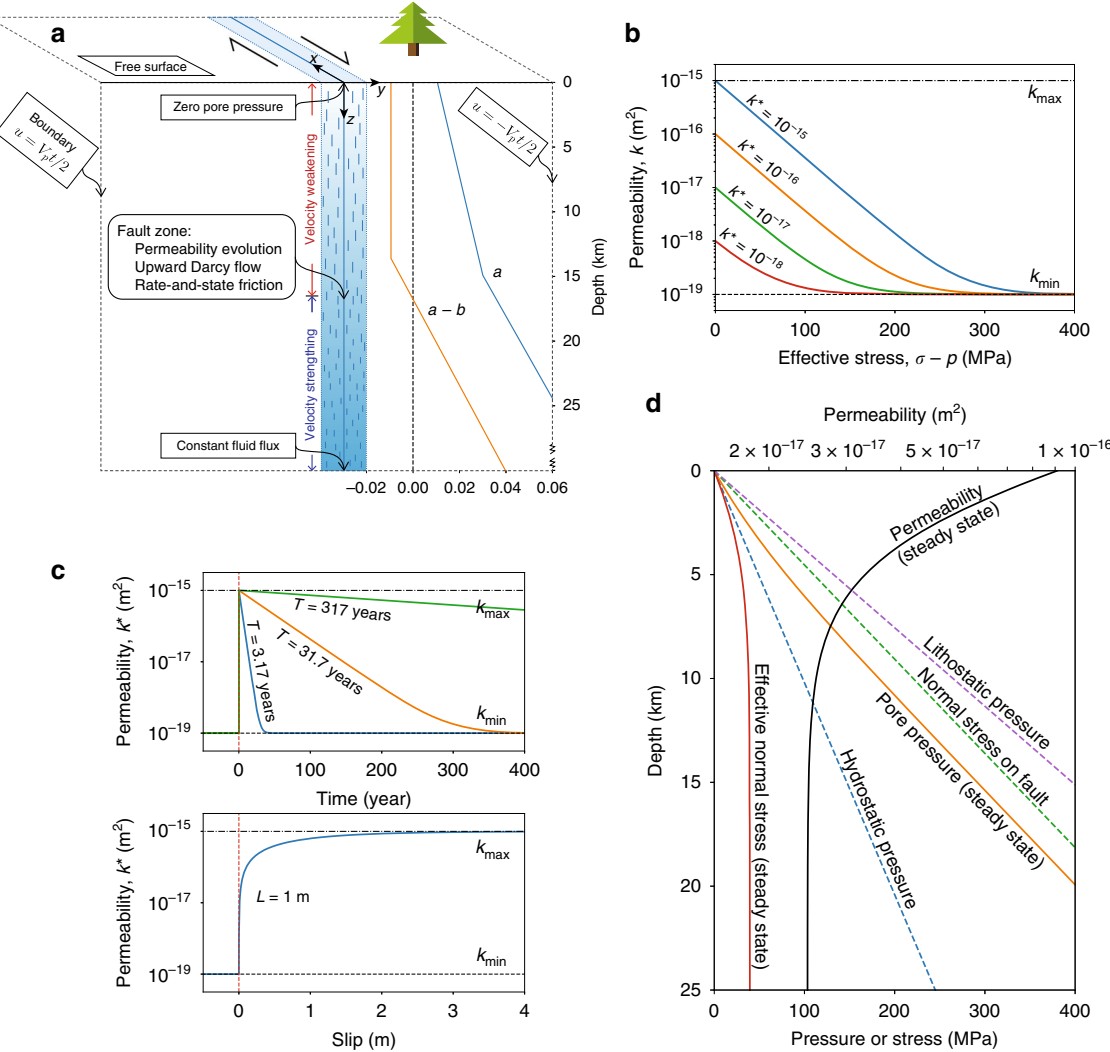

**Fig. 1 Model setting. a** Strike-slip earthquake sequence simulations in a linear elastic solid with rate-and-state friction, fault zone fluid transport, and pore pressure evolution model; distributions of rate-and-state $a$ and $a - b$ shown on right. **b** Permeability decreases with increasing effective normal stress, shown for different $k^*$, a reference permeability that **c** decreases over healing time scale $T$ and increases during coseismic slip over slip distance $L$. **d** Distribution of permeability, pore pressure, and effective normal stress for steady upward fluid flux using the permeability model in **b**, **c**. Note how effective stress becomes independent of depth. Shown for steady flux $q_0 = 3 \times 10^{-9}$ m/s, $\sigma^* = 30$ MPa, $k_{min} = 10^{-19}$ m², $k_{max} = 10^{-15}$ m², $L = 1$ m, $T = 3.17$ years.

shown in Fig. 1b, where $\sigma^* = 30$ MPa is a stress-sensitivity parameter determined by experiments[13,19,20,34,40] (typically of order 10 MPa), $k_{min} = 10^{-19}$ m² is a minimum bound on permeability, and $k^*$ is a reference permeability that is discussed below. Typically $k_{min}$ is set to zero in fitting experimental data, but keeping $k_{min}$ finite is useful for numerical purposes (the very low $k_{min}$ we use plays little role in the system behavior).

Permeability also evolves due to a range of mechanical and chemical processes[35–37,39,48,49]. Here we introduce an idealization that captures two fundamental processes: permeability increase with slip and permeability reduction from healing and sealing processes over longer time scales. The simplest linear evolution equation capturing these processes is

$$\frac{\partial k^*}{\partial t} = -\frac{V}{L}(k^* - k_{max}) - \frac{1}{T}(k^* - k_{min}), \quad (4)$$

where $V$ is slip velocity. The first term on the right side describes permeability increase toward maximum permeability $k_{max} = 10^{-15}$ m² (based on laboratory and in situ measurements[13,19,20,34,49,50]) over slip distance $L = 1$ m, and the second term describes permeability decrease toward minimum permeability $k_{min} = 10^{-19}$ m² over

time scale $T$; see Fig. 1c. This simple parameterization introduces a minimal number of model parameters, making it ideally suited for identification of fundamental effects and quantification of those effects in terms of dimensionless parameters. Note the use of $k^*$ instead of $k$ in Eq. (4); the direct dependence of $k$ on $\sigma - p$ is captured by utilizing the evolving reference permeability $k^*$ from Eq. (4) in Eq. (3). Our model formulation neglects changes in pore volume fraction, pore compressibility, and storage (all of which are likely much smaller than permeability changes[49]), as well as the pressurization that comes from inelastic compaction of pores[46,47], all of which should be added in future studies. Additionally neglected is the temperature (and hence depth) dependence of the time scale $T$ from an Arrhenius thermal activation rate factor for chemical sealing processes like pressure solution[35,37].

The time scale $T$ is poorly constrained, owing to the complexity of processes controlling healing and sealing. Predictions from pressure solution and crack sealing kinetics suggest time scales ranging from days to thousands of years[35,37], and in situ permeability estimates following the 2008 Wenchuan earthquake show healing of the shallow fault within a year[49]. Likewise,

**Table 1 Model parameters and associated references for parameter values.**

| Parameter | Symbol | Value |
|---|---|---|
| **Material** | | |
| Domain dimensions | $L_y, L_z$ | 500 km |
| Plate loading velocity | $V_p$ | $10^{-9}$ m s$^{-1}$ |
| Shear modulus | $\mu$ | 32.4 GPa |
| **Rate-and-state friction** | | |
| Direct and state evolution effect parameters[68,69] | $a, b$ | see Fig. 1a |
| Reference velocity[68] | $V_0$ | $10^{-6}$ m s$^{-1}$ |
| Reference friction coefficient[68] | $f_0$ | 0.6 |
| State evolution distance | $d_c$ | 2 mm |
| Radiation damping coefficient[41,68] | $\eta_{rad}$ | 4.68 MPa s m$^{-1}$ |
| **Fluid transport** | | |
| Gravity | $g$ | 9.8 m s$^{-2}$ |
| Fluid density[70] | $\rho$ | 1000 kg m$^{-3}$ |
| Pore volume fraction[13,19,20,34] | $n$ | 0.01 |
| Fluid viscosity[70] | $\eta$ | $10^{-4}$ Pa s |
| Fluid plus pore compressibility[3] | $\beta$ | $10^{-9}$ Pa$^{-1}$ |
| Imposed fluid flux[12,16] | $q_0$ | $3 \times 10^{-9}$ m s$^{-1}$ (except as noted) |
| **Permeability evolution** | | |
| Stress sensitivity parameter[13,19,20,34,40] | $\sigma^*$ | 30 MPa |
| Permeability enhancement evolution distance | $L$ | 1 m |
| Healing/sealing time scale[35-37,49] | $T$ | $10^8$ s $\approx$ 3.17 years (except as noted) |
| Minimum permeability[13,19,20,34,40,42,49] | $k_{min}$ | $10^{-19}$ m$^2$ |
| Maximum permeability[13,19,20,34,40,42,49] | $k_{max}$ | $10^{-15}$ m$^2$ |

high-temperature laboratory experiments demonstrate that hydrothermal reactions can dramatically reduce permeability, with estimated time scales at mid-seismogenic zone temperatures of a few to a few tens of years[36]. We set $T = 3.17$ years in our featured model but also explore models with alternative choices of $T$ varying over several orders of magnitude.

Also poorly constrained is the permeability enhancement distance $L$, as our simplified evolution equation is an attempt to parameterize complex processes like cracking and yielding within the damage zone from stress concentrations at the rupture tip, dilatancy during shearing of the fault core and slip surface, and unclogging of pores and disruption of grain contacts. We select $L$ so that the steady-state permeability curve (Fig. 1) takes on values broadly consistent with available constraints[13,17–19,49].

First consider steady sliding at plate rate $V_p = 10^{-9}$ m s$^{-1}$. Equation (4) yields a steady-state $k^* = (k_{max}V_p/L + k_{min}/T)/(V_p/L + 1/T) \approx k_{max}/(1 + L/V_pT)$, reflecting a competition between permeability increase from sliding and decrease from healing/sealing. The approximate form, valid for sufficiently small $k_{min}$, highlights the dimensionless parameter $V_pT/L$ that quantifies the relative efficiencies of healing/sealing and permeability enhancement. We then insert this steady state $k^*$ into Eq. (3) and solve Darcy's law (Eq. (2)) for $p$ and $k$, assuming $q = q_0$. This provides the distributions of pore pressure, effective stress, and permeability shown in Fig. 1d. As Rice[21] first showed, the nonlinear dependence of $k$ on $\sigma - p$, under steady flux conditions, creates a pore pressure distribution that transitions from hydrostatic near the surface to tracking the fault normal stress gradient below a few kilometers depth (for representative values of $\sigma^*$). Hence, the effective stress distribution becomes independent of depth over most of the seismogenic zone.

**Fault valving simulation**. We use the steady-state distribution of effective stress in Fig. 1d, held constant for all time, in a reference earthquake sequence simulation. We compare this to a fault valving simulation in which $p$ and $k$ evolve in time following the equations presented above. Results are shown in Figs. 2–4, Supplementary Figs. 1 and 2, and Supplementary Movie 1.

The reference simulation (Fig. 2, top row) has periodic earthquakes that rupture the entire seismogenic zone, and during the interseismic period there is minimal change in the locking depth (i.e., the transition from relatively steady sliding at the plate rate at depth to the locked seismogenic zone). In contrast, the fault valving simulation features more complex phenomena that include fluid-driven aseismic slip and swarm-like seismicity. To explain these phenomena, we divide the earthquake cycle into four phases, labeled 1–4 in Figs. 2 (bottom row) and 3, which show slip velocity and other fields over the earthquake cycle starting after a large earthquake that spans the seismogenic zone and increases its permeability. Time histories of fields at select depths are provided in Supplementary Fig. 1, and Supplementary Movie 1 provides a visualization of the results. During phase 1, the fault discharges fluids from the high permeability seismogenic zone, decreasing overpressure and increasing effective stress. The transition to phase 2 occurs after 5–10 years, when healing/sealing has reduced the seismogenic zone permeability. Influx from depth builds overpressure, which weakens the fault and initiates a fluid-driven aseismic slip front that migrates upward from 20 to 13 km depth over 15 years. Aseismic slip increases permeability, allowing fluid overpressure to advance upward and weaken the fault. Elastic stress transfer also facilitates slip migration, as pointed out by Bhattacharya and Viesca[8]. When this overpressure and aseismic slip front penetrates some distance into the VW seismogenic zone, it nucleates a small earthquake that ruptures 13–18 km depth. In phase 3, overpressure and aseismic slip continue to advance upward, stalling at 8 km after about a decade. Simultaneously, a second aseismic slip and overpressure pulse develops at about 20 km and migrates upward, nucleating a larger earthquake at 12 km depth that ruptures between 8 and 19 km depth. Phase 4 marks the transition to swarm-like seismicity, featuring many relatively small earthquakes that migrate upward following the fluid overpressure pulse as it ascends through the seismogenic zone (Fig. 4). This culminates in the nucleation of a large, surface-breaking rupture. Then this general cycle, with some variations (Supplementary Fig. 2), begins anew.

What controls characteristics of the fault valving process, like changes in overpressure, variations in flux, and propagation rates of the fluid-driven aseismic slip front? We performed a limited parameter-space study varying the healing/sealing time $T$, which controls the duration of depressurization. A key dimensionless parameter is the ratio of $T$ to the recurrence interval of large earthquakes. Models with $T$ comparable to or greater than the earthquake recurrence interval (Supplementary Figs. 3–6, $T = 31.7$ and 317 years) show reduced or even negligible fault valving behavior, as the fault remains a high permeability pathway throughout the earthquake cycle. Models with $T$ much shorter than the recurrence interval (Fig. 5 and Supplementary Movie 2, $T = 0.317$ years) also have reduced overpressure cycling in the seismogenic zone but do exhibit quasi-periodic slow slip events that are spontaneously generated at the base of the seismogenic zone. These slow slip events are the fluid-driven aseismic slip fronts identified in Figs. 2 and 3 for $T = 3.17$ years, but the shorter $T$ increases the rate at which they are generated so that many occur between each earthquake. Furthermore, decreasing $T$ increases the propagation rate of the aseismic slip fronts (Fig. 6).

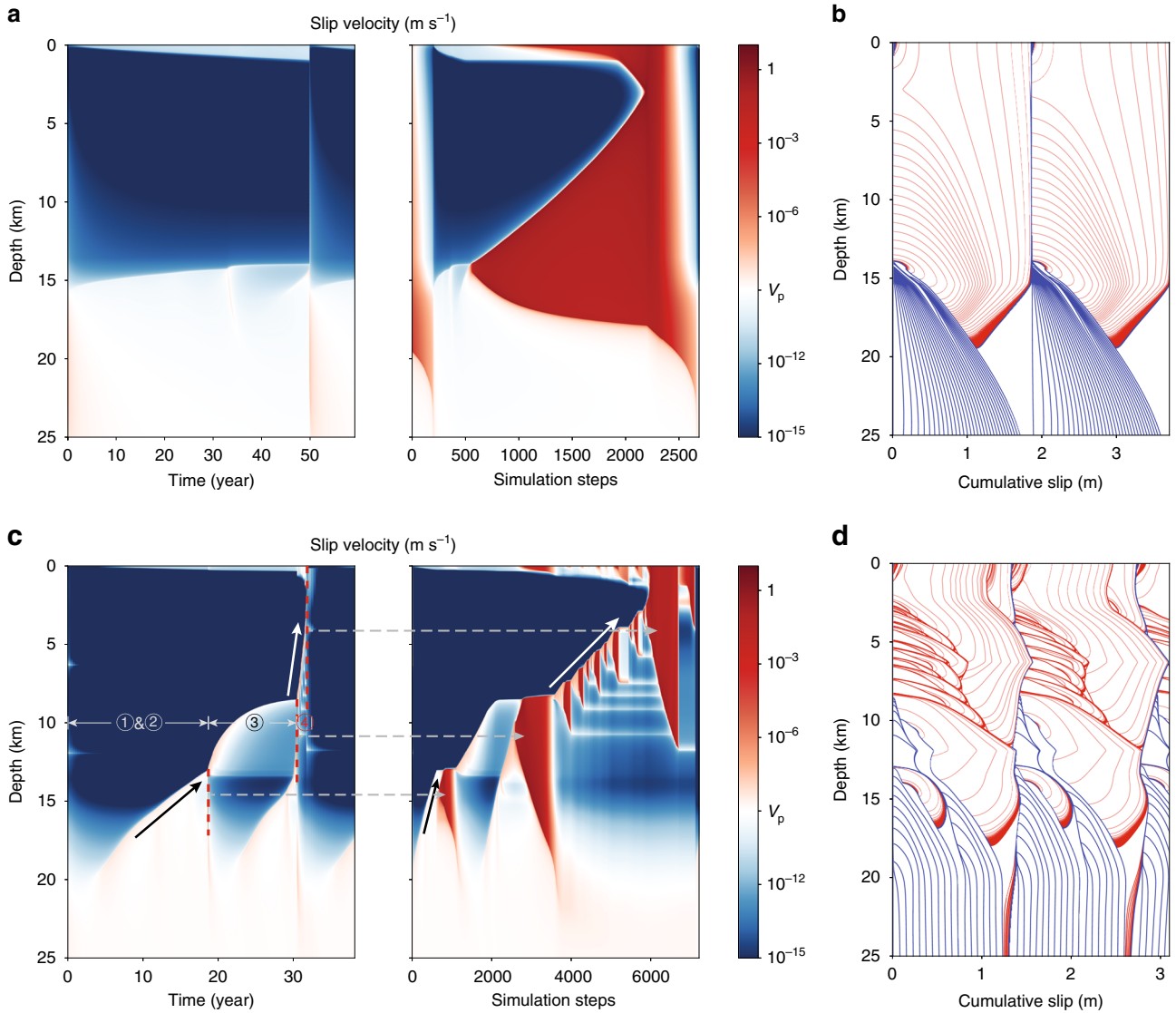

**Fig. 2 Evolution of slip velocity and slip. a**, **b** Reference model; **c**, **d** fault valving model. Slip contours in **b**, **d** are plotted in the coseismic period in red every 1 s and in interseismic period in blue every 1.5 years. In the fault valving model, fluid-driven aseismic slip fronts (with accompanying overpressure, Fig. 3a, b) emerge from the base of the seismogenic zone (e.g., black arrow), migrating upward and nucleating earthquakes. The continued ascent of the overpressure pulse triggers swarm seismicity in the mid-seismogenic zone (white arrow, also Fig. 4). Slip at Earth's surface that accompanies deeper seismic events is an artifact of the quasi-dynamic elastic approximation and should be ignored.

Returning to the featured model in Figs. 2 and 3, we quantitatively explain the fault valving characteristics. The maximum flux following a large earthquake can be estimated from Darcy's law (Eq. (2)) with a pressure gradient bounded approximately by the fault normal stress gradient and the maximum permeability: $q_{max} \approx (k_{max}/\eta)(d\sigma/dz - \rho g) \sim 10^{-7}$ m s$^{-1}$ (the actual pressure gradient is controlled by the ability of the seismogenic zone to pressurize during the late interseismic period, which depends on influx, minimum permeability, and storage). The depressurization rate of the seismogenic zone follows from integrating Eq. (1) across the seismogenic zone of width $H$ (and neglecting spatial variations in depressurization rate), with outflux equated to $q_{max}$ and negligible influx at depth: $dp/dt \approx -q_{max}/(n\beta H) \sim 10$ MPa year$^{-1}$. The actual depressurization rate is smaller than this upper bound, due to somewhat lower pressure gradient, permeability, and outflux. The depressurization duration is controlled by $T$, leading to an overall pressure drop of $q_{max}T/(n\beta H) \sim 10$ MPa.

## Discussion

Our modeling predicts two phenomena that can be compared to observations: fluid-driven aseismic slip and swarm-like seismicity, both arising as overpressure pulses migrate upward along the fault. Fluid-driven aseismic slip might be observable in geodetic data as a progressive decrease of plate coupling or an ascending locking depth, though trade-offs in geodetic inversions might make this hard to resolve. Furthermore, if the deep part of the fault has heterogeneous frictional properties, microseismicity might accompany aseismic slip, as shown by Jiang and Lapusta[51].

Analysis of decadal-scale deformation data (from Global Positioning System, leveling, and tide gauges) in the Cascadia subduction zone provides evidence for a gradual unlocking of the transition zone between the locked seismogenic zone and the deeper region of episodic tremor and slip[52]. The data are consistent with a model in which deep aseismic slip migrates up-dip at a rate of 30–120 m year$^{-1}$, not too dissimilar to our example fault valving simulation in Figs. 2 and 3. That model, with $T = 3.17$ years, has a

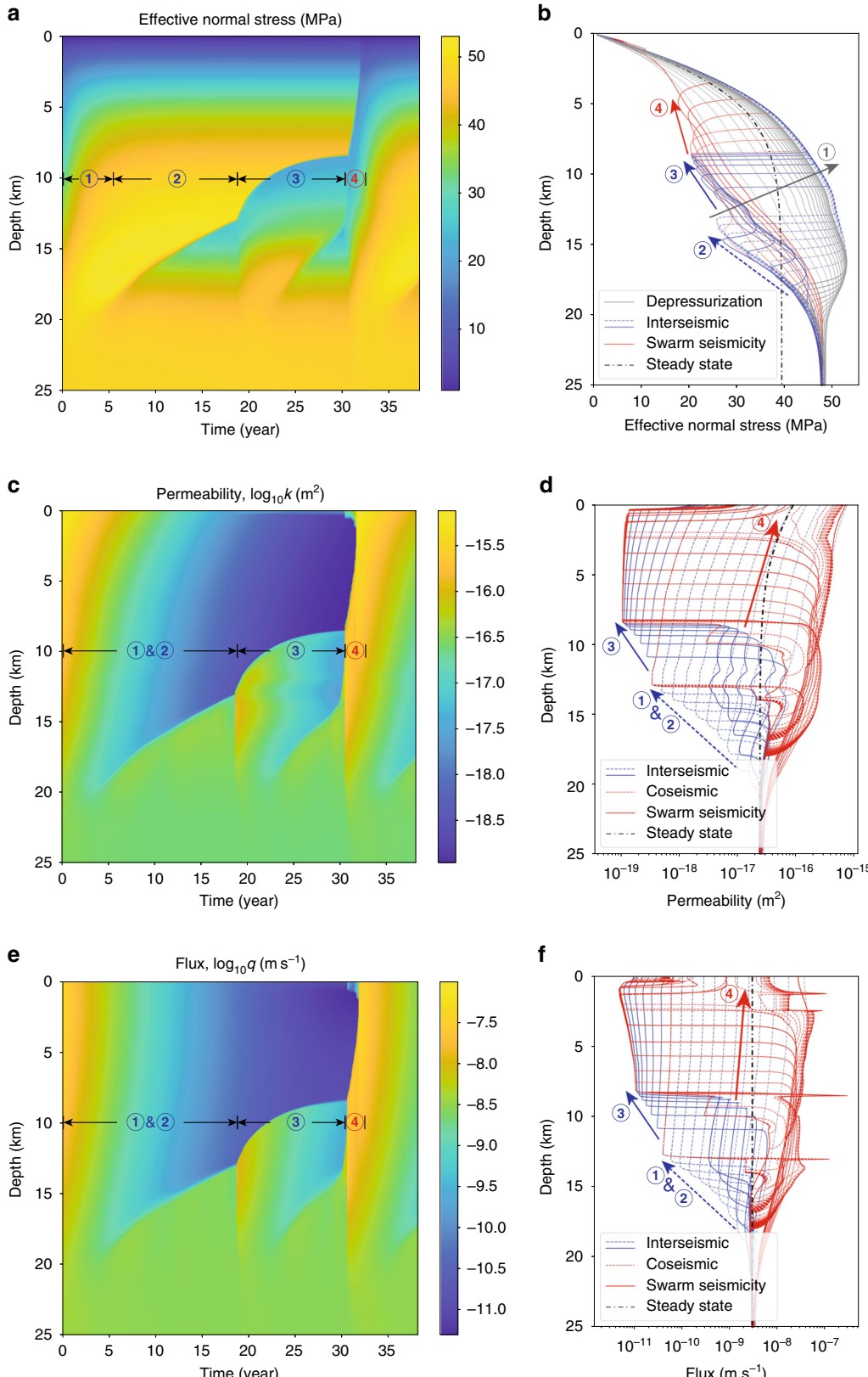

migration rate of 380 m year⁻¹, and we find that increasing $T$ decreases the migration rate (e.g., $T = 31.7$ years has a migration rate of 120 m year⁻¹, Fig. 6).

Fluid-driven aseismic slip might also help explain slow slip events that occur in subduction zones[53–57] and at the base of the seismogenic zone in the Parkfield section of the San Andreas

fault[58]. These tectonic settings are associated with high pore pressures, arising in part from metamorphic reactions like serpentinite dehydration that liberate fluids at near-lithostatic pressures. In these various settings, slow slip can migrate both along-strike and up- and down-dip. The fluid-driven aseismic slip phenomenon that we identified could equally well occur in the

**Fig. 3 Fault valving behavior.** Evolution of **a**, **b** effective normal stress; **c**, **d** permeability; and **e**, **f** fluid flux. Various phases are labeled with circled numbers in **a**, **c**, **e**, and with line color and dashes in **b**, **d**, **f**; the steady-state solution is shown in dashed black lines. During phase 1, the seismogenic zone depressurizes (gray lines in **b**, every 0.5 years) following a large earthquake. Permeability and flux within the seismogenic zone decrease during phase 2; simultaneously, an overpressure pulse emerges from about 20 km depth and migrates upward, with aseismic slip increasing permeability and allowing influx of fluids (dashed blue lines, every 1.5 years). After a small earthquake, this overpressure pulse continues upward in phase 3 and a second fluid-driven aseismic slip front and overpressure pulse emerges from depth (solid blue lines, every 1.5 years). A larger earthquake marks the transition to phase 4, where swarm-like seismicity accompanies the ascending overpressure pulse (solid red lines, 0.2 years). Changes during earthquakes are shown in dashed red lines, every 1 s, in **d**, **f**.

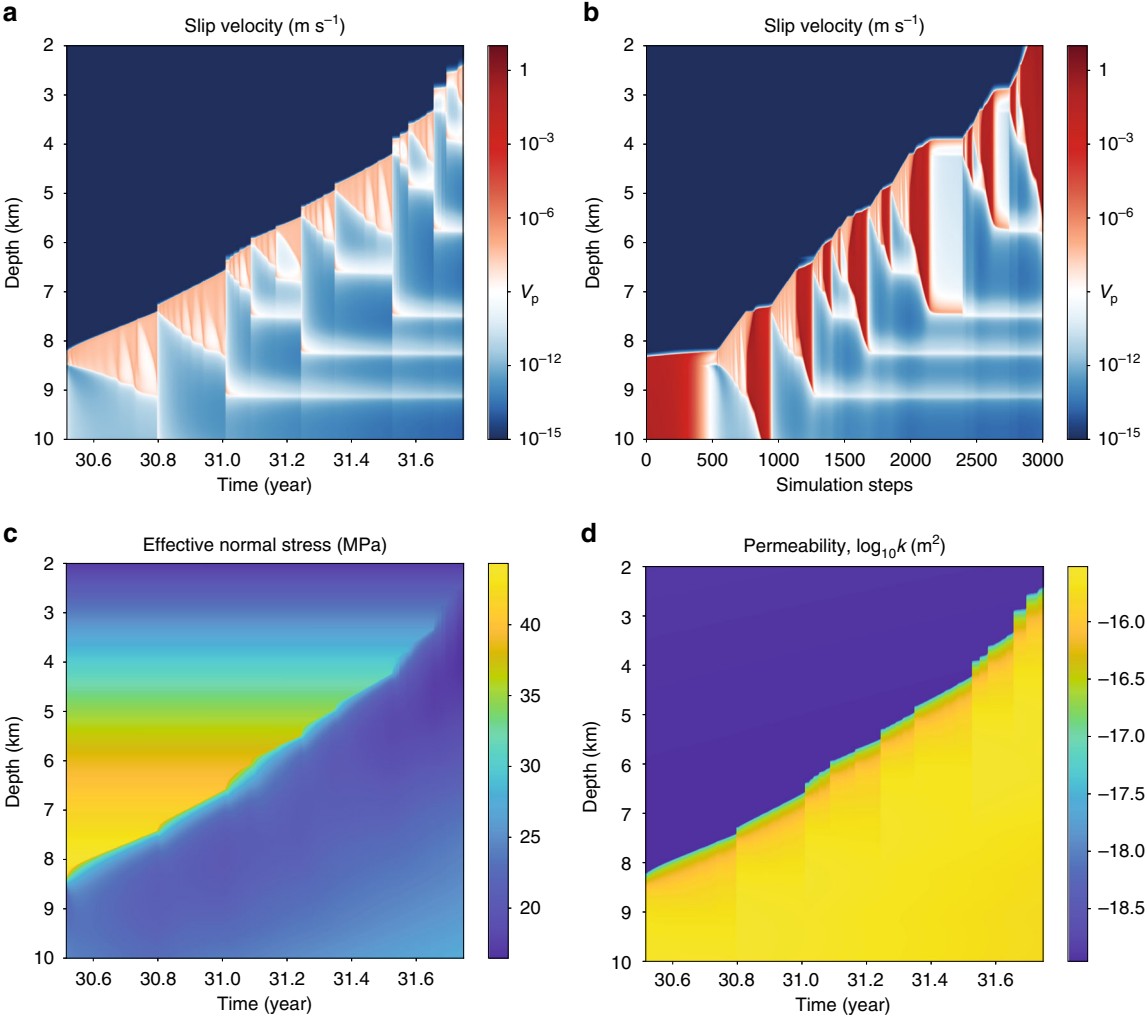

**Fig. 4 Zoomed-in view of swarm seismicity triggered as the overpressure pulse ascends through the mid-seismogenic zone.** Evolution of **a**, **b** slip velocity; **c** effective normal stress; and **d** permeability.

horizontal direction, if there exist lateral variations in frictional properties, fluid production rate, slip velocity, or simply nonlinear dynamics that give rise to spatial variations in pore pressure and associated horizontal pressure gradients. However, the migration rates in our simulations are much slower than observed slow slip propagation rates, and additional simulations exploring higher fluid fluxes $q_0$, lower effective stresses, and other parameter variations are required to test the viability of this hypothesized explanation for slow slip events. That said, our model with $T = 0.317$ years (Fig. 5) does produce quasi-periodic slip events with duration of about 1 year, repeating every few years, and with slip of a few cm. This is similar to so-called long-term slow slip events that have been observed at the base of the seismogenic zone in Japan, New Zealand, and elsewhere[59–62]. The short healing/

sealing times required to produce these slow slip events are arguably consistent with the high temperatures expected at these depths.

We also suggest that fluid-driven aseismic slip might play a role in induced seismicity and reservoir geomechanics, where many observations indicate pore pressure and/or stress communication across large distances at time scales far shorter than expected from pore pressure diffusion with typical or measured hydraulic diffusivities[8,29,63,64]. Our study builds on recent work[8,65] highlighting how the coupling between aseismic slip and pore pressure diffusion can rapidly transmit pressure changes. The nonlinearities accounted for in our simulations, specifically the permeability increase from slip and reductions in effective stress, make this process even more efficient.

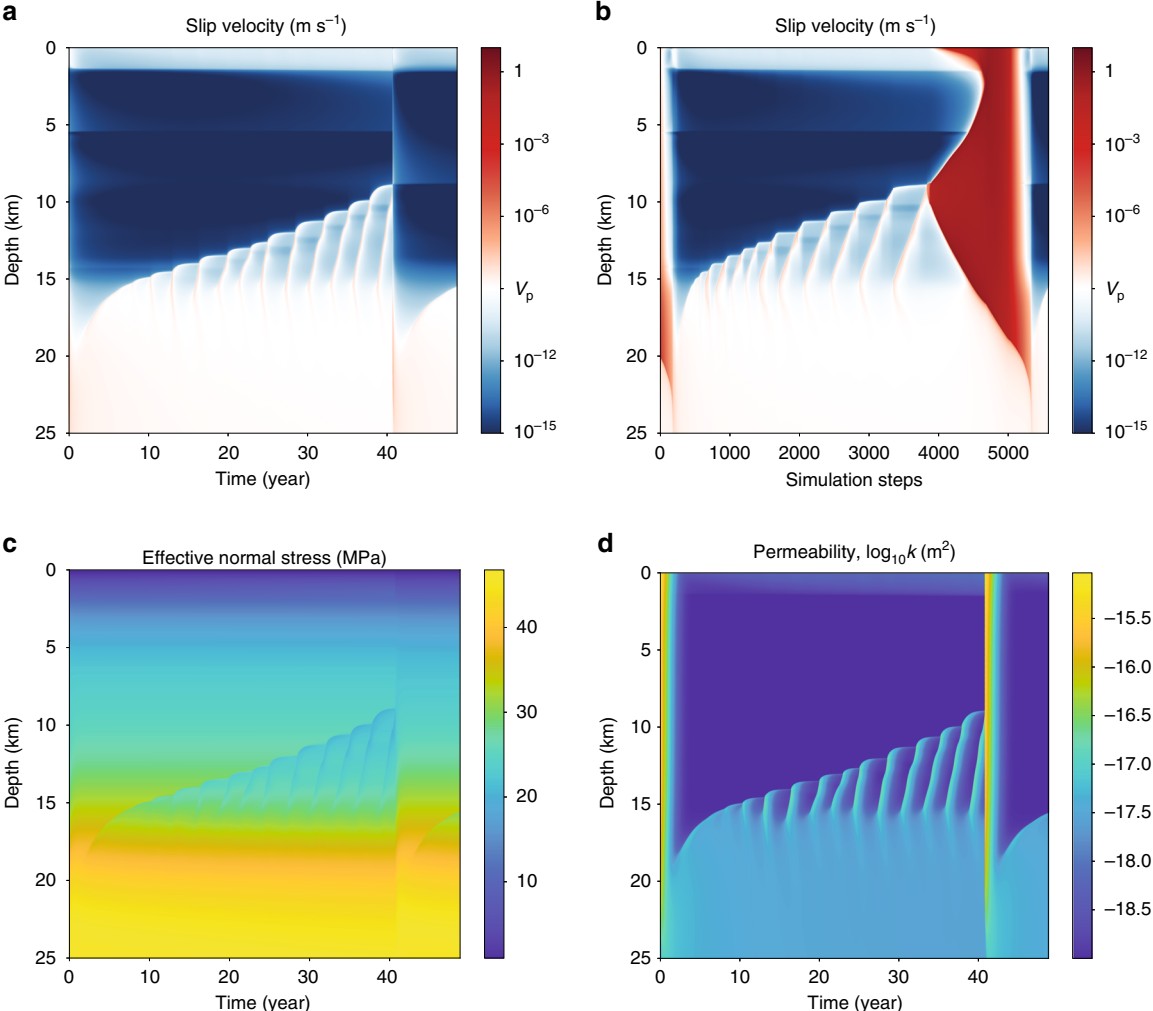

**Fig. 5 Quasi-periodic aseismic slip pulses.** Evolution of **a**, **b** slip velocity; **c** effective normal stress; and **d** permeability in the fault valving simulation with $T = 0.317$ years (and $q_0 = 3.3 \times 10^{-10}$ m s$^{-1}$ rather than $3 \times 10^{-9}$ m s$^{-1}$, with flux chosen to give similar effective stress at depth to $T = 3.17$ year model). Quasi-periodic fluid-driven aseismic slip pulses, akin to long-term slow slip events in subduction zones, are spontaneously generated at the base of the seismogenic zone. The reference simulation for this case (not shown) is similar to the reference simulation for $T = 3.17$ years in Fig. 2a, b and has only periodic, large earthquakes.

The second phenomenon in our simulations, swarm seismicity, is commonly associated with regions of active fluid transport, such as volcanic fields and geothermal sites[24,26,27,29]. Our simulations demonstrate that overpressure pulses can ascend in concert with swarm-like seismic events that, in addition to or instead of aseismic slip, transiently enhance permeability to allow continued overpressure advancement. In our simulations, swarm seismicity requires rate-weakening friction and sufficiently small nucleation length (e.g., from sufficiently small state evolution distance and/or high effective stress; for specific parameters influencing nucleation length, see ref. [66]). Further studies exploring a broader range of parameters, particularly fluid fluxes $q_0$ and dependence on frictional parameters like $a - b$ and $d_c$ (see "Methods"), are required to match seismicity migration rates observed in specific sequences.

Overall, we have demonstrated the viability of fault valving in an earthquake sequence model that accounts for permeability evolution and fault zone fluid transport. Predicted changes in fault strength from cyclic variations in pore pressure are substantial (~10–20 MPa) and perhaps even larger than those from changes in friction coefficient. We have also shown how fluids facilitate the propagation of aseismic slip fronts and transmission of pore

pressure changes at relatively fast rates. The modeling framework we have introduced here can be applied to a wide range of problems, including tectonic earthquake sequences, slow slip and creep transients, earthquake swarms, and induced seismicity.

## Methods

**Friction and elasticity.** The numerical method for the friction and elasticity problem is identical to that in several previous publications[67,68] using fourth-order summation-by-parts finite differences for spatial discretization and adaptive Runge–Kutta time stepping. The frictional strength of the fault is determined by rate-and-state friction with an aging law:

$$f(\psi, V) = a \sinh^{-1}\left(\frac{V}{2V_0} e^{\psi/a}\right), \tag{5}$$

$$\frac{\partial \psi}{\partial t} = \frac{bV_0}{d_c}\left(e^{(f_0 - \psi)/b} - \frac{V}{V_0}\right), \tag{6}$$

where $\psi$ is the state variable, $V$ is the slip velocity, $a$ is the direct effect parameter, $V_0$ is the reference velocity, $b$ is the state evolution effect parameter, $d_c$ is the state evolution distance, and $f_0$ is the reference friction coefficient for steady sliding at $V_0$.

The antiplane displacement $u$ (in the $x$ direction) is governed by the static equilibrium equation and Hooke's law:

$$\frac{\partial \sigma_{xy}^{qs}}{\partial y} + \frac{\partial \sigma_{xz}^{qs}}{\partial z} = 0, \quad \sigma_{xy}^{qs} = \mu \frac{\partial u}{\partial y}, \quad \sigma_{xz}^{qs} = \mu \frac{\partial u}{\partial z}, \tag{7}$$

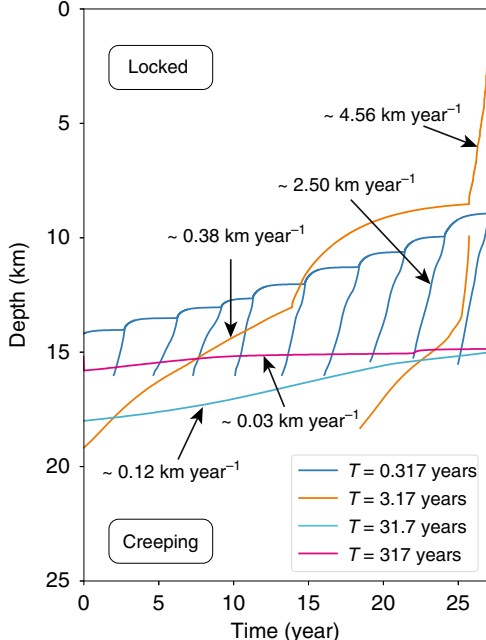

**Fig. 6 Comparison of propagation rates of fluid-driven aseismic slip and overpressure fronts in fault valving simulations with various $T$.**
Decreasing $T$ leads to faster propagation rates. The $T = 0.317$ year model uses $q_0 = 3.3 \times 10^{-10}$ m s$^{-1}$ rather than $3 \times 10^{-9}$ m s$^{-1}$, as in the other models.

where $\sigma_{xy}^{qs}$ and $\sigma_{xz}^{qs}$ are the shear stresses in this quasi-static problem and $\mu$ is the shear modulus. Symmetry conditions across the fault ($y = 0$) are used to solve the problem on one side of the fault only, in the domain $0 \leq y \leq L_y$, $0 \leq z \leq L_z$.

Fault frictional strength is equated to the shear stress on the fault, which is the sum of the quasi-static shear stress and a radiation damping term:

$$\sigma_{xy}^{qs} - \eta_{\text{rad}} V = f(V, \psi)(\sigma - p) \quad \text{on} \quad y = 0, \tag{8}$$

where $\eta_{\text{rad}}$ is the radiation damping coefficient[41], $f(V, \psi)$ is the friction coefficient, $\sigma$ is the total normal stress on the fault, and $p$ is the pore pressure. Slip $\delta$ is defined via $\partial\delta/\partial t = V$. In solving the elasticity problem for $u$, slip is prescribed on the fault, tectonic loading displacement is prescribed on the side boundary, and traction-free conditions are prescribed on the top and bottom boundaries:

$$u(0, z, t) = \frac{\delta}{2}, \quad u(L_y, z, t) = \frac{V_p t}{2}, \quad \sigma_{xz}^{qs}(y, 0, t) = 0, \quad \sigma_{xz}^{qs}(y, L_z, t) = 0. \tag{9}$$

The pore pressure diffusion equation is discretized using fourth-order summation-by-parts finite differences, like the elasticity equation.

**Time stepping**. Here we explain the time-stepping method. An adaptive Runge–Kutta method is used to update $\delta$ and $\psi$ with variable time steps $\Delta t$, as in previous work[67,68]. The difference here is that we must simultaneously solve the pore pressure diffusion and permeability evolution equations to update $p$, $k^*$, and $k$. This is done using operator splitting at the Runge–Kutta stage level, with backward Euler used for the pore pressure diffusion equation. More details are provided below, with the algorithm explained using forward Euler instead of the explicit Runge–Kutta method for simplicity.

All dependent variables ($\delta$, $\psi$, $p$, $k^*$, $k$) are known at time $t$. Then we update from time $t$ to $t + \Delta t$ following the procedure below:

1. Solve the equilibrium Eq. (7) for $u(t)$ and calculate $\sigma_{xy}^{qs}(t)$ on the fault.
2. Solve Eq. (8) for velocity $V(t)$ using $p(t)$ when evaluating fault strength.
3. Update $\psi(t + \Delta t)$, $\delta(t + \Delta t)$, and $k^*(t + \Delta t)$ explicitly, e.g.,

$$k^*(t + \Delta t) = k^*(t) + \Delta t \left( -\frac{V}{L}(k^*(t) - k_{\max}) - \frac{1}{T}(k^*(t) - k_{\min}) \right). \tag{10}$$

4. Implicitly update $p(t + \Delta t)$ and $k(t + \Delta t)$:

$$n\beta \frac{p(t + \Delta t) - p(t)}{\Delta t} = \frac{\partial}{\partial z}\left[ \frac{k(t + \Delta t)}{\eta}\left( \frac{\partial p(t + \Delta t)}{\partial z} - \rho g \right) \right], \tag{11}$$

$$k(t + \Delta t) = k_{\min} + (k^*(t + \Delta t) - k_{\min})e^{-(\sigma - p(t + \Delta t))/\sigma^*}. \tag{12}$$

The nonlinear system is solved using fixed-point iteration:

Convergence is declared when the difference of successive updates to $p'$ drops below a tolerance. While spatial operators are written here for the continuum problem, the numerical solution is obtained for the spatially discretized problem where inverting the operator means solving a linear system with appropriate boundary conditions.

1: $p' \leftarrow p(t)$
2: **while** not converged **do**
3: $\quad k' \leftarrow k_{\min} + (k^*(t + \Delta t) - k_{\min})e^{-(\sigma - p')/\sigma^*}$
4: $\quad p' \leftarrow \left( \frac{n\beta}{\Delta t} - \frac{\partial}{\partial z}\frac{k'}{\eta}\frac{\partial}{\partial z} \right)^{-1} \left( \frac{n\beta p(t)}{\Delta t} - \frac{\partial}{\partial z}\frac{k' \rho g}{\eta} \right)$
5: **end while**
6: $p(t + \Delta t) \leftarrow p'$, $k(t + \Delta t) \leftarrow k'$

**Model parameters**. The parameters used in this study are shown in Table 1. The depth distribution of $a$ and $a - b$ is similar to Allison and Dunham[68] and other previous modeling studies[41] and is based on laboratory experiments[69] and an assumed geotherm. The state evolution distance $d_c$, which is proportional to the earthquake nucleation length, is chosen to be as small as possible while balancing computational cost.

### Data availability
The simulation data in this study are available in Open Science Framework: https://doi.org/10.17605/OSF.IO/9YGRP.

### Code availability
All simulations were performed in the open-source code Scycle, available at https://bitbucket.org/kallison/scycle.

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

## Acknowledgements

This research was supported by the National Science Foundation (EAR-1947448) and the Southern California Earthquake Center (Contribution No. 9931). SCEC is funded by NSF Cooperative Agreement EAR-1600087 & USGS Cooperative Agreement G17AC00047.

## Author contributions

W.Z. added pore pressure diffusion to Scycle (with assistance from K.L.A. and Y.Y. on implementation, debugging, and checkpointing) and ran all simulations. E.M.D. designed the study. W.Z. and E.M.D. jointly interpreted results and wrote the paper, with all authors provided feedback.

## Competing interests

The authors declare no competing interests.
