## [Peer Review File · Nature Communications]

REVIEWER COMMENTS

Reviewer #1 (Remarks to the Author):

The authors present an intriguing modeling study of the importance of the two-way coupling between fault slip and the hydraulic properties of a fault zone in producing fluid-induced, aseismically-driven unlocking of the seismogenic locked zone. In particular, the model provides an elegant and quantitative explanation for the updip migration of deep creep during the interseismic period and gradual shallowing of the unlocking depth prior to seismic slip.

While modeling the role of fluids in determining the dynamics of fault-slip is not new, this study provides the first model of spontaneously fluid-controlled earthquake cycles on tectonically driven faults while explicitly accounting for both permeability enhancement during slip and permeability reduction during the interseismic period. This allows the simulations to capture the cycle of fault-valving in detail in the context of tectonically loaded faults.

Additionally, the potentially wide-applicability of models of this type to study fluid induced aseismic slip and its role in promoting fault instability across a diverse range of geological settings makes this work of broad interest and definitely worthy of publication in Nature Comm. after some revisions.

In what follows, I present a list of my concerns which the authors might find useful to address:

Major concern:

- 1) The effect of the permeability enhancement and healing parameters: One important aspect of the model is the competition between permeability enhancement due to slip and reduction due to fault healing. The authors present simulations where the time scale for permeability enhancement at steady-state (L/V_{lp}) and the healing period (T) are equal. While a detailed exploration of the parameter space might be out of the scope of the study, it would have been nice to see end-member simulations where $L/V_{lp} \ll T$ or $L/V_{lp} \gg T$. It would be interesting to see how these parameter choices might affect both the recurrence interval and the updip migration rate of deep aseismic slip.

Particularly, given that the authors provide a justification of their model based on comparisons of their simulated updip migration rate of aseismic slip to crustal scale observations, it would be interesting to learn what parameters control this property of their simulations.

Additionally, these end-member simulations might lead to very different seismic-aseismic transitions. For example, in the case that healing is very effective, one might produce much larger pore-pressures at depths shallower than the 17 km rheological transition. This, while weakening the fault might also lead to longer times to instability (since nucleation length scales increase with decreasing effective normal stress) and in turn influence recurrence intervals. Also, the rate of propagation of fluid-induced aseismic slip could also be sensitive to pore-pressure levels. It would be great if the authors could provide some information on

such end-member behavior of the model. It might be fine to include these simulations as supplementary material for reference to the interested reader.

Minor concerns:

- 1) Comparisons between Figures 2 and 3: I am slightly confused while comparing Figures 2d and 3d. While the seismic slip regimes in the locked portion of the fault are invisible in Figure 2d are invisible (due to the time scale of the x-axis?), the resultant permeability enhancements in the cycles are clearly visible in Figure 3d. While I understand that this has to do with the time required to see the effects of healing but it would still be illuminating to see the slip rate variation snapshots in Figure 3 along with the present panels on effective normal stress (3a), permeability (3c) and fluid-flux (3e). Alternatively, maybe the authors could come up with some other way of plotting this info that might allow one to see the spatial inter-relationship between slip rate and permeability and pore-pressure evolution in the same figure.
- 2) Figure 1a: Is the value of a constant at depths shallower than 17 km or decreasing with depth as shown in the Figure? Table 1 does not mention the values of a and b used in the simulations.
- 3) “As Rice (1992) first showed, ..., becomes independent of depth.”: Maybe modify the last sentence to say ‘becomes independent of depth at relatively shallow depths for small values of σ^* ’?
- 4) Caption of Figure 3: Should “Red and blue curves plotted every 1 yr and 4 yr in (a) and every 2 s and 5 yr in (c) and (e)” be “Red and blue curves plotted every 1 s and 4 yr in (a) and every 2 s and 5 yr in (c) and (e)”?
- 5) “The fluid-driven aseismic slip phenomenon that we identified could equally well occur in the horizontal direction, if there exist lateral variations in fluid production rate, fault locking depth, or simply nonlinear dynamics that give rise to spatial variations in pore pressure and associated horizontal pressure gradients.”

By horizontal variations in fault locking depth, do the authors mean slip-rate gradients in the horizontal direction? That might indeed lead to large horizontal variations in permeability.
- 6) It might be worthwhile to mention why the mechanical compression of pores and microfractures is cited as the cause behind variations in permeability yet porosity (n) is independent of depth?
- 7) Might be useful to mention how the fluid-viscosity value is chosen?

Pathikrit Bhattacharya

The manuscript uses novel numerical modeling with combined evolution of fault slip and permeability to demonstrate the feasibility of the fault-valving hypothesis. The idea is that, under the steady flux of fluids at depth, the fault not only would be overpressured but also, due to slip- and healing-induced variations in permeability, would cycle between states of higher and lower pore pressure at depth that would significantly affect the fault strength, promote rapid migration of aseismic slip from the bottom of the fault, and significantly affect earthquake nucleation. This is a valuable study that brings into focus a previously hypothesized but somewhat forgotten notion. It provides a potential explanation to significant, near-lithostatic, fluid overpressure hypothesized, based on a range of observations, at the bottom of seismogenic portions of subduction zones and at the roots of some strike-slip faults as referenced in the manuscript. The model adopts understandable simplifications to highlight the effects, although some aspects would benefit from additional explanation. The manuscript is well-written and will be thought provoking for a wide audience interested in earthquake source processes. I recommend that it be published, after a revision to account for the following comments.

1. The model incorporates a number of simplifications, which is completely understandable. Most of them are well discussed in the manuscript. The pore fluid pressure evolution equation (1) incorporates fluid diffusion along the fault (in the depth direction) but seems to ignore the fault-normal diffusion, which is never discussed. Yet the pore pressure gradient in the off-fault direction could be just as large - if not larger - than along the fault. While the model in the manuscript seems to assume significant fluid overpressure everywhere in the bulk (the steady state shown in Fig. 1d), in the case of the San Andreas fault referenced in the manuscript, there is no evidence for the significant pore fluid overpressure in the entire bulk surrounding the fault (say, 5-10 km away) at the seismogenic depths. This would mean that the near-lithostatic pore pressures predicted by the model are confined to the damage zone around the fault while pore pressure several kilometers away from the fault is near-hydrostatic. That would create significant fault-normal pore pressure gradient and hence fluid flow. Furthermore, even if the entire bulk has the steady-state overpressure shown in Fig. 1d, the fluctuations of pore fluid pressure within the fault shown by the modeling would still create fluid flow in the fault-normal direction. It would be helpful to the reader to (a) explicitly state in the manuscript that the steady state of Fig. 1d is assumed everywhere in the bulk, (b) discuss what it means to the applicability of the model to the San Andreas Fault and similar settings, and (c) mention that the off-fault diffusion is ignored and discuss the consequences of that.

2. It is not clear what motivates the choices of the slip distance of 1 m, time scale of 31.7 years, and permeability constants that govern the evolution of permeability in equations (3-4). Are they chosen to maximize the effects presented in the manuscript? Are they based on laboratory studies? Are they plausible for natural faults? It would be important to explain, as the values are simply stated in the manuscript. As referenced in the manuscript, the fault-valving hypothesis was already supported by simplified modeling in previous studies (references 8, 16, 17), so the question is not so much whether it is plausible at all, but rather whether it can happen for realistic fault properties/modeling.

3. If lab/field studies point to a range of potential choices for parameters in comment 2, or if these parameters are not constrained, then it would be helpful to add a small parameter study that shows the significance of these choices. It is stated in the manuscript that such exploration is "beyond the scope of this study" but the current presentation seems to be based on two simulations of a model that is highly simplified geometrically. Such fault simplifications are specifically useful to allow for a tractable parameter exploration. The parameter study need not be exhaustive but it would help to have some idea of the ranges of parameters that lead to fault valving vs. not.

4. The manuscript presents a well-referenced and informative discussion of issues related to pore fluid pressure and its effect on fault slip. The presented results appear to be relevant to a range of potential observations, including significant fluid overpressure of subductions zones at depth and upward migration of fault unlocking (typically called decoupling). The manuscript also states that: "Fluid-driven aseismic slip might also help explain slow slip events that occur in subduction zones 35-39." However, the episodic slow slip events from subduction zones propagate orders of magnitude faster horizontally than the speed of creeping fronts shown in the presented modeling, so it seems quite unlikely that this could be a direct explanation; that is why a range of other physical mechanisms has been proposed. Perhaps the authors mean something else; it would be good to clarify.

5. The quasi-dynamic depth-variable fault modeling used in the study was introduced by Tse and Rice (JGR, 1986) and Rice (JGR, 1993), and it would be appropriate to cite those works. Incidentally, it would be helpful for the authors to explain the value of the radiation damping coefficient in Table 1, as it seems different from Rice (1993), for example.

Reviewer #3 (Remarks to the Author):

This manuscript presents the results of a thoughtful and interesting, focused study that rigorously couples rate and state friction (RSF), rupture dynamics, and fluid transport/pressure diffusion. It builds on the “classic” paper from Rice (1992) that illustrated the potential for mantle-derived fluids to weaken vertical faults (e.g., the San Andreas) over much of the seismogenic depth range, given some simple assumptions about stress-dependence of fault permeability and a specified basal flux driven by an assumed mantle devolatilization rate.

The authors extend the ideas of Rice (1992) in some important ways, most notably in assessing the effect of that same upward mantle flux on earthquake cycles. The most interesting result, in my view, is the effect that fluid pressure propagation has during the interseismic period, namely through : (1) gradual shallowing of the locked zone; and (2) predicted precursory or interseismic creep below the seismogenic zone. The paper is novel, and represents a clear advance through the incorporation of fluid flow and RSF.

Having said that, I think it’s probably borderline call whether the work is appropriate for Nature Communications. I’ve outlined a handful of specific suggestions below that I think would help convey the broader interest and significance of the work, and thus appeal to a wider audience. If these can be addressed (and they are fairly addressable), I would definitely recommend publication in Nature Communications, as the work has potentially very broad application & impact, and should garner substantial interest. These center on (a) presentation of the key / new findings; and (b) drawing a stronger and more direct connection to the kinds of observations referenced at the end of the abstract (points #2, 4, 5, and 6 below).

- 1) In the second paragraph of the main text, the passage beginning with “Fault permeability is typically...”: this is only somewhat true. Observations of mature faults in crystalline rock (or other lithified/competent/brittle rocks) generally show this, but it isn’t universal. In thinking about the broader potential application of the results reported by the authors, this clarification is important. There are useful reviews of fault permeability architecture by Caine et al. (1996) and Bense et al. (2010) that would provide a framework for this clarification.
- 2) I understand that for the sake of modeling simplicity, and to explore basic systematic behaviors, it’s necessary to fix some parameters, or at least limit the number of free parameters. However, it seems to me that the choice of (b-a) is one that could ultimately govern a lot of the system’s behavior. Hence, I’d like to see (i) a better justification of that choice (top of p. 4), as well as the choice of D_c ; and (ii) some discussion of the sensitivity of the model results to this choice. For example, the amount and duration of aseismic precursory slip might be substantially diminished if (b-a) were larger than 0.001, such that the seismogenic zone could fail unstably more readily? The same logic likely applies to the choice of (b-a) below 17 km and the fact that it increases linearly to the base of the model.

- 3) The first paragraph on p. 5 is somewhat confusing. Strictly speaking, from a hydrological point of view, “flow” occurs in response to pressure gradients. That redistribution of fluid mass does, ultimately, redistribute pressure. But I don’t think it’s totally accurate to say that the “upward flow...leads to overpressure”. It would be better to say that the prescribed upward flux at the model base leads to overpressure.
- 4) Top of page 12 – these two paragraphs highlight the main and novel contribution of the work in my estimation. I think this needs to be brought out more strongly and emphasized both in the text as a primary result, and in the figures (see next suggestion). To some extent this also ties to previous comment #2, which speaks to how robust this result is, and/or how strongly this behavior hinges on parameter choices.
- 5) Related to #4, I found that the graphics do show the key result, but the fundamental new insight might be emphasized with an additional or alternate graphic so it isn’t lost in the details. Specifically, why not distill the result with a plot showing the depth of locking as $f(\text{time})$ for a few different parameter choices? This would illustrate both the sensitivity noted above, and bring forward the important idea that the upward-percolating fluids “erode” the locked zone from below. Likewise, I think a plot of the amount of creep (or slip rate at a couple of distinct depths) as $f(t)$ for a couple of different depths would be useful. The bottom line is that there are some important and really interesting outcomes here, and I think they are discernible but perhaps a bit lost in the details of figure 2, because there is a lot shown there (and because color contour plots are not as effective in conveying quantitative information as line plots).
- 6) I almost hate to say it, but a cartoon emphasizing the observations discussed in the last section of the paper (bottom of p. 12 through p. 13) might be helpful. The ideas and observations introduced in this section varied – ranging from gradual unlocking (this *is* explicitly reproduced in the models), to SSE (this *is not*...so a better discussion of how this might be linked to the models is needed, and particularly how the much faster rates of “unzipping”, on the order of km/d rather than tens of m/yr, would – or could – be explained by the models), to transient creep events on the SAF (which is linked only speculatively to the modeling results). Broadly speaking, I think this section of the paper is a key element in engaging a broader audience and demonstrating the role that fluid migration may play in driving interseismic /precursory fault slip – so should be strengthened through more specificity and some direct comparisons of rates, timescales, etc...

Minor comments:

- a) Abstract, lines 4-6. Seems more appropriate to say “...quantify *the effects of* fault valving through 2D...friction, *in the face of upward flux of fluids from the mantle* ~~upward Darcy flow along a permeable fault zone~~, and permeability evolution.” [see also comment #3 above]
- b) Abstract, line 9: replace “shearing of the fault zone” with “slip”?
- c) P. 2, line 1: end of abstract: isn’t “fault unlocking in the late interseismic period” basically “precursory slow [or aseismic] slip”? It seems clearer and more emphatic to say so if that’s the case.

- d) P. 2: the rock mechanics community usually uses μ for friction coefficient. That may be problematic because the same symbol is also used for fluid viscosity, but worth thinking about the audience here.
- e) P. 2 end of first paragraph of main text: dilatancy hardening has also been explored – not just thermal pressurization.
- f) End of that same paragraph and elsewhere – it’s awkward and not very useful to use “etc.”. Suggest deleting, or replacing with specific list of items or at least a clearer articulation of the kinds of things that are tuned. Might be worth citing Liu & Rice for this statement also.
- g) P. 2, 4th line from bottom: “Flow along faults creates pressure gradients” isn’t quite accurate – see comment #3 above. Flow occurs in response to pressure gradients and ultimately serves to dissipate them.
- h) Second to last line: delete the word “perhaps”
- i) Last line: replace “the high rates” simply with “to”
- j) P. 3, first line: overpressure doesn’t result from compaction and lithification. It’s the other way around – those changes in the rock or sediment occur in response to flow of fluids and escape of overpressure. The water is pressurized by mechanical loading (burial + tectonic stress), and then flows away (i.e. dewatering).
- k) Last paragraph of intro: Warren Smith (2019) just published a paper on valving related to SSE in New Zealand that should be cited here.
- l) P. 4 first full paragraph: seems important to either justify the argument that country rock is “impermeable” or discuss briefly how fluid leakage into the wall rock might impact the results.
- m) P. 4 bottom: see “f” above, use of “etc.”
- n) P. 5 top: the fluxes observed on continental plate boundary faults are highly focused at springs and other discharge points rather than uniformly distributed along the fault length. This would imply that at depth, the flux (per m of fault length) is smaller, and this then is “collected” somehow along the path from deep crust to surface.
- o) P. 5, below eq (3): 10^{-19} could be considered a pretty high number for k of lower crustal rock, even faults. When effective stress is high (i.e. the “minimum bound”), a number 10-100 times lower would be reasonable (see Lockner and his group’s work on Nojiema and SAF rock). The numbers from Manning & Ingebritsen (1999) are long-term averages and likely represent mainly flux during phases of elevated pore pressure driven by metamorphic fluid release.
- p) P. 5 after equation 4: likewise, the choice of upper bound on k is probably low. Observations of post-seismic fluid discharge, along with modeling studies constrained by data indicating thermal or geochemical advection, suggest k values that are ~10-1000 times higher. It would be interesting to see if this would drive more rapid migration of aseismic slip fronts (see point #6 above – particularly as related to the question of explaining SSE which “unzip” much faster than a few tens of m/yr).
- q) P. 6 and more generally: how is normal stress defined? Is it simply a Poisson effect, or is tectonic loading included?
- r) P. 8, statement “...leads to an earthquake.” should be referenced.

- s) P. 8: would be useful to make clear in the main text whether k increases only in the part of the fault that slips, or if there is some damage assumed in a halo or region that extends some distance from the actively slipping patches ends.
- t) P. 12: It's a bit nuanced, and maybe doesn't matter hugely, but generally speaking, if the fault is healing hydrologically, it implies a reduction in porosity (cracks or intergranular). This same process should somehow be connected to frictional healing through the parameter b . By that logic, one might expect the value of RSF parameter b (high values indicating greater healing) to map at least qualitatively to the rate of permeability increase. Likewise where b is small (i.e. greater depths in the model space as shown in Fig 1), contact area and porosity don't evolve or heal much, so this ought to map to decreased rates of change of k .
- u) P. 13, middle paragraph, last sentence: this would also occur if there are lateral changes in $(b-a)$ which may be as, or more, likely than variations in fluid production rate.
- v) Same paragraph, last sentence: doesn't this also fundamentally require a transition from negative to positive $(b-a)$? I think that's what the models are showing – that aseismic failure takes place in regions that were previously locked, but were never going to nucleate instability anyway because they are rate weakening?

We thank the reviewers for your valuable feedback and suggestions, which has led us to perform additional simulations and revisions to the manuscript. In particular, at the suggestion of one of the reviewers, we explored the sensitivity of model results to the choice of frictional state evolution distance. By using a smaller, and hence more realistic, state evolution distance we reduced the nucleation length and discovered an additional phenomenon: swarm-like seismic events. We feel that this new discovery is quite important and relevant to our study, so we made substantial revisions to our manuscript to describe it, along with the fluid-driven aseismic slip that we had focused on in the original manuscript. Furthermore, we also found that with shorter healing/sealing time, the model produces aseismic slip pulses that have properties which are quite similar to long-term slow slip events in subduction zones.

We also want to emphasize that our study is of generic processes and phenomena arising from the coupling between fluids and frictional slip, rather than trying to be the most realistic model of the San Andreas Fault, Alpine Fault, or any other specific fault. At the start of the Model section, we added some text to explain this: “The purpose of this study is to introduce a quantitative simulation framework in which to explore the two-way coupling between fluid transport, pore pressure evolution, and fault slip over the earthquake cycle. Our focus is on the processes and phenomena that arise from this coupling, in a generic sense. We do this in the context of a quasi-dynamic 2-D antiplane shear model of a vertical strike-slip fault in a uniform elastic half-space (Fig. 1a), the classic idealization for investigation of processes controlling earthquake sequences and aseismic slip. While parameter choices are chosen to be reasonably representative of continental strike-slip plate boundary settings, we are not attempting to model any specific fault or earthquake sequence. Furthermore, it is possible that key findings might be relevant to other tectonic settings like subduction zones.” We hope this clarifies the intent of the study, and that the manuscript is evaluated within the context of this intent.

Below we describe the changes that were made in response to reviewers' comments:

Reviewer 1:

The authors present an intriguing modeling study of the importance of the two-way coupling between fault slip and the hydraulic properties of a fault zone in producing fluid-induced, aseismically-driven unlocking of the seismogenic locked zone. In particular, the model provides an elegant and quantitative explanation for the updip migration of deep creep during the interseismic period and gradual shallowing of the unlocking depth prior to seismic slip.

While modeling the role of fluids in determining the dynamics of fault-slip is not new, this study provides the first model of spontaneously fluid-controlled earthquake cycles on tectonically driven faults while explicitly accounting for both permeability enhancement during slip and permeability reduction during the interseismic period. This allows the simulations to capture the cycle of fault-valving in detail in the context of tectonically loaded faults.

Additionally, the potentially wide-applicability of models of this type to study fluid induced aseismic slip and its role in promoting fault instability across a diverse range of geological settings makes this work of broad interest and definitely worthy of publication in Nature Comm. after some revisions.

Author response: We thank the reviewer for recognizing the broad applicability of this modeling framework and for placing the work into context.

In what follows, I present a list of my concerns which the authors might find useful to address:

Major concern:

1) The effect of the permeability enhancement and healing parameters: One important aspect of the model is the competition between permeability enhancement due to slip and reduction due to fault healing. The authors present simulations where the time scale for permeability enhancement at steady-state (L/V_{pl}) and the healing period (T) are equal. While a detailed exploration of the parameter space might be out of the scope of the study, it would have been nice to see end-member simulations where $L/V_{pl} \ll T$ or $L/V_{pl} \gg T$. It would be interesting to see how these parameter choices might affect both the recurrence interval and the updip migration rate of deep aseismic slip.

Particularly, given that the authors provide a justification of their model based on comparisons of their simulated updip migration rate of aseismic slip to crustal scale observations, it would be interesting to learn what parameters control this property of their simulations.

Additionally, these end-member simulations might lead to very different seismic-aseismic transitions. For example, in the case that healing is very effective, one might produce much larger pore-pressures at depths shallower than the 17 km rheological transition. This, while weakening the fault might also lead to longer times to instability (since nucleation length scales increase with decreasing effective normal stress) and in turn influence recurrence intervals. Also, the rate of propagation of fluid-induced aseismic slip could also be sensitive to pore-pressure levels. It would be great if the authors could provide some information on such end-member behavior of the model. It might be fine to include these simulations as supplementary material for reference to the interested reader.

Author response: We now present simulations for four healing times: $T = 0.317, 3.17, 31.7,$ and 317 yr (the latter two in Supplementary Figures), and discuss briefly in the main text how the system behavior depends upon T : “A key dimensionless parameter is the ratio of T to the recurrence interval of large earthquakes. Models with T comparable to or greater than the earthquake recurrence interval (Supplementary Figs. 3-6, $T = 31.7$ and 317 yr) show reduced or even negligible fault valving behavior, as the fault remains a high permeability pathway throughout the earthquake cycle. Models with T much shorter than the recurrence interval (Fig.

5 and Supplementary Movie 2, $T = 0.317$ yr) also have reduced overpressure cycling in the seismogenic zone, but do exhibit quasi-periodic slow slip events that are spontaneously generated at the base of the seismogenic zone.” We did not investigate recurrence intervals because with the switch to small state evolution distance, we obtain more complex event sequences rather than characteristic events that always rupture the entire seismogenic zone. We feel that the fluid-driven aseismic slip and swarm seismicity phenomena are the most important ones to focus on. We found that the propagation rate of the aseismic slip fronts are relatively insensitive to state evolution distance (the new simulations in the paper with $dc = 2$ mm have rates that are similar to the ones in the previous simulations with $dc = 10$ mm), but are quite sensitive to T . To show this in detail we added Figure 6. We suspect, but have not confirmed in simulations, that the propagation rate also depends on the influx q_0 . However, given the length limits of this journal, we felt that exploring a broader range of q_0 was out of the scope of what could be done.

Minor concerns:

1) Comparisons between Figures 2 and 3: I am slightly confused while comparing Figures 2d and 3d. While the seismic slip regimes in the locked portion of the fault are invisible in Figure 2d are invisible (due to the time scale of the x-axis?), the resultant permeability enhancements in the cycles are clearly visible in Figure 3d. While I understand that this has to do with the time required to see the effects of healing but it would still be illuminating to see the slip rate variation snapshots in Figure 3 along with the present panels on effective normal stress (3a), permeability (3c) and fluid-flux (3e). Alternatively, maybe the authors could come up with some other way of plotting this info that might allow one to see the spatial inter-relationship between slip rate and permeability and pore-pressure evolution in the same figure.

Author response: Thanks for pointing out the challenges of understanding our results from the figures in the original manuscript. We have extensively revised the figures and text to explain solution behavior in detail. To show what happens during the coseismic period, we added plots of slip velocity as a function of time step in addition to time (e.g., Fig. 2a and c). These highlight and emphasize the coseismic phase. We also added Supplementary Movie 1 that provides a movie of slip velocity and other fields on the fault, which provides much insight into the coupled evolution of the fields.

2) Figure 1a: Is the value of a constant at depths shallower than 17 km or decreasing with depth as shown in the Figure? Table 1 does not mention the values of a and b used in the simulations.

Author response: The values of a and $a - b$ are plotted in Figure 1a. We added a reference to this figure in Table 1. The values of a and b as a function of temperature are from Allison and Dunham (2018), which are based on fitting laboratory data (Blanpied, 1991) and extrapolation assuming linear dependence of a on temperature. These are then converted to

profiles with depth based on an assumed geotherm. The chosen geotherm and distribution of a and b are identical to a currently unpublished study by Allison and Dunham, and utilize a cooler geotherm than in Allison and Dunham (2018). This is why the velocity-weakening to velocity-strengthening transition is deeper than in their study.

3) “As Rice (1992) first showed, ..., becomes independent of depth.”: Maybe modify the last sentence to say ‘becomes independent of depth at relatively shallow depths for small values of σ^* ’?

Author response: We modified this brief summary of the Rice model in the revised manuscript.

4) Caption of Figure 3: Should “Red and blue curves plotted every 1 yr and 4 yr in (a) and every 2 s and 5 yr in (c) and (e)” be “Red and blue curves plotted every 1 s and 4 yr in (a) and every 2 s and 5 yr in (c) and (e)”?

Author response: The caption was correct, with different contour spacing used for overpressure and the other fields. This is because overpressure has negligible evolution over the coseismic period, whereas permeability and hence flux change dramatically during the coseismic period. In the revised figures, we have added additional colors and line formats to highlight various processes. In addition, we have expanded the description in the caption to make it clear to the reader what to focus on in the figures.

5) “The fluid-driven aseismic slip phenomenon that we identified could equally well occur in the horizontal direction, if there exist lateral variations in fluid production rate, fault locking depth, or simply nonlinear dynamics that give rise to spatial variations in pore pressure and associated horizontal pressure gradients.”

By horizontal variations in fault locking depth, do the authors mean slip-rate gradients in the horizontal direction? That might indeed lead to large horizontal variations in permeability.

Author response: Yes, exactly. We changed the term fault locking depth to slip velocity to clarify. We anticipate that real faults have a complex history of seismic and aseismic slip, which is variable in space and time, and hence the permeability and overpressure distribution will be quite heterogeneous. This will create horizontal pressure gradients and hence horizontal fluid flow. The same feedback between aseismic slip, permeability enhancement, inflow of fluid, pressurization, etc., might then occur horizontally.

6) It might be worthwhile to mention why the mechanical compression of pores and microfractures is cited as the cause behind variations in permeability yet porosity (n) is independent of depth?

Author response: Porosity (and perhaps also compressibility) decreases with depth, which means storage will depend on depth. However, in this study we decided to focus exclusively on the effect of permeability and therefore we made porosity and compressibility independent of

depth for simplicity. With the permeability evolution law we used in this paper, the behavior of the fault system is already quite complex and our objective was not to make the most realistic simulations, but to isolate the effect of what we suspect is the most important dynamic fluid transport property. In future work, we intend to account for additional processes like inelastic porosity changes (i.e., creep compaction) that other authors like Sleep and Blanpied (1992) have identified as being important for fluid migration, especially at depths where viscoelastic pore closure time is shorter than other time scales in the problem like the earthquake recurrence interval.

7) Might be useful to mention how the fluid-viscosity value is chosen?

Author response: Fluid viscosity at room temperature (20°) is about 10^{-3} Pa s and decreases to 10^{-4} Pa s at 260°, so we choose it to 10^{-4} Pa s in our simulation considering the higher temperature in the crust. The uncertainty of viscosity is much smaller than permeability which varies by several orders of magnitude across different studies. This comment, as well as others, prompted us to provide more discussion of parameter choices in the text. We also added references for parameter choices to Table 1.

Reviewer 2:

The manuscript uses novel numerical modeling with combined evolution of fault slip and permeability to demonstrate the feasibility of the fault-valving hypothesis. The idea is that, under the steady flux of fluids at depth, the fault not only would be overpressured but also, due to slip- and healing-induced variations in permeability, would cycle between states of higher and lower pore pressure at depth that would significantly affect the fault strength, promote rapid migration of aseismic slip from the bottom of the fault, and significantly affect earthquake nucleation. This is a valuable study that brings into focus a previously hypothesized but somewhat forgotten notion. It provides a potential explanation to significant, near-lithostatic, fluid overpressure hypothesized, based on a range of observations, at the bottom of seismogenic portions of subduction zones and at the roots of some strike-slip faults as referenced in the manuscript. The model adopts understandable simplifications to highlight the effects, although some aspects would benefit from additional explanation. The manuscript is well-written and will be thought provoking for a wide audience interested in earthquake source processes. I recommend that it be published, after a revision to account for the following comments.

1. The model incorporates a number of simplifications, which is completely understandable. Most of them are well discussed in the manuscript. The pore fluid pressure evolution equation (1) incorporates fluid diffusion along the fault (in the depth direction) but seems to ignore the fault-normal diffusion, which is never discussed. Yet the pore pressure gradient in the off-fault direction could be just as large - if not larger - than along the fault. While the model in the manuscript seems to assume significant fluid overpressure everywhere in the bulk (the steady

state shown in Fig. 1d), in the case of the San Andreas fault referenced in the manuscript, there is no evidence for the significant pore fluid overpressure in the entire bulk surrounding the fault (say, 5-10 km away) at the seismogenic depths. This would mean that the near-lithostatic pore pressures predicted by the model are confined to the damage zone around the fault while pore pressure several kilometers away from the fault is near-hydrostatic. That would create significant fault-normal pore pressure gradient and hence fluid flow. Furthermore, even if the entire bulk has the steady-state overpressure shown in Fig. 1d, the fluctuations of pore fluid pressure within the fault shown by the modeling would still create fluid flow in the fault-normal direction. It would be helpful to the reader to (a) explicitly state in the manuscript that the steady state of Fig. 1d is assumed everywhere in the bulk, (b) discuss what it means to the applicability of the model to the San Andreas Fault and similar settings, and (c) mention that the off-fault diffusion is ignored and discuss the consequences of that.

Author response: You've brought up several important points. First, though, we would like to clarify several things. Please interpret this as a generic fault model rather than a model of either the San Andreas or Alpine Faults (or any other specific fault). We discuss these faults because they are some of the few strike-slip faults for which there are estimates of fluid fluxes, but in both cases there are far more complex hydrologic processes occurring in the bulk. The San Andreas does not have localized fluid upwelling around it, but instead there is a more broadly distributed geochemical signature of deeply sourced fluids. And this signature, and possible overpressure, is confined to one side of the fault, suggesting that the fault acts as a barrier to flow. Fluid is probably transported along or at least guided by the fault, but it's undoubtedly not nearly as simple as the uniform width high permeability channel in our model. The Alpine Fault apparently involves downward and lateral transport of meteoric fluids (i.e., rainfall) driven by topographic gradients into the fault around the brittle-ductile transition prior to ascent upward along the fault. Given that each of these scenarios would be complex to parameterize, we chose to focus on phenomena that emerge from permeability evolution and coupling between fluid transport and slip in a more idealized setting. With this background, then, we chose to focus on the most common permeability structure observed around faults (i.e., high permeability damage zone with anisotropy that promotes along-fault flow, e.g., Faulkner and Rutter, 2001, and other references in the text). Modeling by Faulkner and Rutter (2001) explored the effect of horizontal pressure gradients and horizontal outflow from overpressured faults, but it seems like the effects of that horizontal transport are reasonably small for most parameter choices. We also neglect processes like poroelastic pressure changes (e.g., Rudnicki and Rice, 2006; Dunham and Rice, 2008) and thermal pressurization (e.g., Rice, 2006) that would lead to large fault-normal pressure gradients on coseismic time scales. Instead, we view our model as being applicable on time scales that exceed that hydraulic diffusion time across the fault (damage) zone width. Of course this assumption breaks down during the coseismic period, but we consider it an acceptable approximation given the benefit of model simplification.

2. It is not clear what motivates the choices of the slip distance of 1 m, time scale of 31.7 years, and permeability constants that govern the evolution of permeability in equations (3-4). Are they chosen to maximize the effects presented in the manuscript? Are they based on laboratory

studies? Are they plausible for natural faults? It would be important to explain, as the values are simply stated in the manuscript. As referenced in the manuscript, the fault-valving hypothesis was already supported by simplified modeling in previous studies (references 8, 16, 17), so the question is not so much whether it is plausible at all, but rather whether it can happen for realistic fault properties/modeling.

Author response: We now provide discussion of parameter choices in the text, and references are also compiled in Table 1. We have done our best to select realistic parameters, but there is considerable uncertainty in the healing/sealing time T and the permeability enhancement distance L . To address uncertainty in T , we now present models for four different values of T .

3. If lab/field studies point to a range of potential choices for parameters in comment 2, or if these parameters are not constrained, then it would be helpful to add a small parameter study that shows the significance of these choices. It is stated in the manuscript that such exploration is “beyond the scope of this study” but the current presentation seems to be based on two simulations of a model that is highly simplified geometrically. Such fault simplifications are specifically useful to allow for a tractable parameter exploration. The parameter study need not be exhaustive but it would help to have some idea of the ranges of parameters that lead to fault valving vs. not.

Author response: The revised manuscript presents results of four simulations having different healing/sealing time T . This parameter appears to exert the primary control on the fault valving behavior as well as the propagation rate of the fluid-driven aseismic slip front. Although we have used a simplified 2D strike slip model, the small nucleation length we now use requires very fine grids for proper numerical resolution, and each simulation must be run through many earthquake cycles to spin-up the system and eliminate the sensitivity to arbitrarily chosen initial conditions. These factors introduce a high computational cost that, together with the space limitations of this journal, prohibit a more thorough parameter-space exploration.

4. The manuscript presents a well-referenced and informative discussion of issues related to pore fluid pressure and its effect on fault slip. The presented results appear to be relevant to a range of potential observations, including significant fluid overpressure of subductions zones at depth and upward migration of fault unlocking (typically called decoupling). The manuscript also states that: “Fluid-driven aseismic slip might also help explain slow slip events that occur in subduction zones 35–39.” However, the episodic slow slip events from subduction zones propagate orders of magnitude faster horizontally than the speed of creeping fronts shown in the presented modeling, so it seems quite unlikely that this could be a direct explanation; that is why a range of other physical mechanisms has been proposed. Perhaps the authors mean something else; it would be good to clarify.

Author response: This is a fair point, and we openly acknowledge in the revised manuscript that the predicted propagation rates of the aseismic slip fronts in our simulations are much slower than in slow slip events. However, our additional simulations exploring a broader range

of healing/sealing times did produce much faster propagation rates than in our original simulation, though rates are still slower than those observed in subduction slow slip (specifically, short-term slow slip events). However, we have not fully explored parameter space, specifically not conditions of near lithostatic pore pressure and (transiently?) high fluid fluxes that arguably characterize the slow slip regions of subduction zones. We suspect there might be parameter choices that produce behaviors similar to observed slow slip events. Our simulations with very short healing time, which might be appropriate to the base of the seismogenic zone in subduction zones, do produce events with properties that are quite similar to long-term slow slip events. While this is very speculative, we do hope the reviewer will indulge us to leave this idea in the manuscript. We hope we have been honest about this in writing, “However, the migration rates in our simulations are much slower than observed slow slip propagation rates, and additional simulations exploring higher fluid fluxes q_0 , lower effective stresses, and other parameter variations are required to test the viability of this hypothesized explanation for slow slip events. That said, our model with $T = 0.317$ yr (Fig. 5) does produce quasi-periodic slip events with duration of about 1 yr, repeating every few years, and with slip of a few cm. This is similar to so-called long-term slow slip events that have been observed in Japan, New Zealand, and elsewhere. The short healing/sealing times required to produce these slow slip events are arguably consistent with the high temperatures expected at these depths.”

5. The quasi-dynamic depth-variable fault modeling used in the study was introduced by Tse and Rice (JGR, 1986) and Rice (JGR, 1993), and it would be appropriate to cite those works. Incidentally, it would be helpful for the authors to explain the value of the radiation damping coefficient in Table 1, as it seems different from Rice (1993), for example.

Author response: We added a citation to Rice (1993) when discussing radiation damping in the Methods section. The value of the radiation damping coefficient is the same as Allison and Dunham (2018) and is chosen as suggested by Rice (1993) to be $\mu/(2*cs)$. We had mistakenly reported μ/cs in Table 1, but have corrected that now. Thanks for catching this.

Reviewer 3:

This manuscript presents the results of a thoughtful and interesting, focused study that rigorously couples rate and state friction (RSF), rupture dynamics, and fluid transport/pressure diffusion. It builds on the “classic” paper from Rice (1992) that illustrated the potential for mantle-derived fluids to weaken vertical faults (e.g., the San Andreas) over much of the seismogenic depth range, given some simple assumptions about stress-dependence of fault permeability and a specified basal flux driven by an assumed mantle devolatilization rate.

The authors extend the ideas of Rice (1992) in some important ways, most notably in assessing the effect of that same upward mantle flux on earthquake cycles. The most interesting result, in my view, is the effect that fluid pressure propagation has during the interseismic period, namely

through : (1) gradual shallowing of the locked zone; and (2) predicted precursory or interseismic creep below the seismogenic zone. The paper is novel, and represents a clear advance through the incorporation of fluid flow and RSF.

Having said that, I think it's probably borderline call whether the work is appropriate for Nature Communications. I've outlined a handful of specific suggestions below that I think would help convey the broader interest and significance of the work, and thus appeal to a wider audience. If these can be addressed (and they are fairly addressable), I would definitely recommend publication in Nature Communications, as the work has potentially very broad application & impact, and should garner substantial interest. These center on (a) presentation of the key / new findings; and (b) drawing a stronger and more direct connection to the kinds of observations referenced at the end of the abstract (points #2, 4, 5, and 6 below).

Author response: Thanks for the constructive suggestions. Your suggestion below to explore different state evolution distances prompted us to run simulations with much smaller d_c , leading us to discover the swarm seismicity phenomenon in addition to the original phenomenon of fluid-driven aseismic slip fronts. We feel that these two phenomena will be quite interesting to the earthquake processes community, and that our extensively revised manuscript will reach the standards of Nature Communications.

1) In the second paragraph of the main text, the passage beginning with "Fault permeability is typically...": this is only somewhat true. Observations of mature faults in crystalline rock (or other lithified/competent/brittle rocks) generally show this, but it isn't universal. In thinking about the broader potential application of the results reported by the authors, this clarification is important. There are useful reviews of fault permeability architecture by Caine et al. (1996) and Bense et al. (2010) that would provide a framework for this clarification.

Author response: Thanks for pointing this out. We have qualified that sentence by adding "For mature faults in crystalline rock" and we added another sentence highlighting the diversity and variations in the fluid transport properties of faults: "The fluid transport properties of fault zones are highly variable, as a consequence of differences in structure, lithology and composition, stress state, and deformation history (Caine et al. (1996) and Bense et al. (2013))"

2) I understand that for the sake of modeling simplicity, and to explore basic systematic behaviors, it's necessary to fix some parameters, or at least limit the number of free parameters. However, it seems to me that the choice of (b-a) is one that could ultimately govern a lot of the system's behavior. Hence, I'd like to see (i) a better justification of that choice (top of p. 4), as well as the choice of D_c ; and (ii) some discussion of the sensitivity of the model results to this choice. For example, the amount and duration of aseismic precursory slip might be substantially diminished if (b-a) were larger than 0.001, such that the seismogenic zone could fail unstably more readily? The same logic likely applies to the choice of (b-a) below 17 km and the fact that it increases linearly to the base of the model.

Author response: The (b-a) profile is similar to Allison and Dunham (2018), and is based on Blanpied et al. (1991), but with a cooler geotherm that places the VW-VS transition deeper than in their study. We used exactly the same (b-a) profile in another, currently unpublished study by Allison and Dunham (except in Kali Allison's PhD thesis), on thermomechanical coupling in earthquake sequence simulations. We follow Allison and Dunham (2018) in extrapolating (b-a) and a to temperatures (i.e., depths) beyond the experimental constraints (which is what leads to the linear increase at the base of the model). We feel that it is beyond the scope of this study to explore alternative distributions of b-a and a. We modified the manuscript (in the Methods section) to provide a brief justification for the choice of (b-a) and a. Regarding state evolution distance, our original simulations used $d_c = 10$ mm, but reviewers' comments prompted us to explore cases with 5 mm and 2 mm. This led us to discover the swarm seismicity phenomenon that is reported in the revised manuscript. We chose to feature the $d_c = 2$ mm simulations in the revised manuscript, as smaller d_c is more consistent with laboratory experiments and existence of small earthquakes. Please note that switching to a smaller d_c does not eliminate fluid-driven aseismic slip, which can even occur in the mid-seismogenic zone (Supplementary Fig. 2a and b, white arrow) for certain stress conditions. We also found that the propagation rate of aseismic slip fronts is relatively insensitive to d_c , but is sensitive to healing/sealing time T , which we explain in the revised manuscript.

Your suggestion that aseismic slip might not happen if the fault is more strongly rate-weakening is interesting, and perhaps this is related to why the ascending overpressure pulse creates swarm seismicity instead of aseismic slip in the mid-seismogenic zone when we use small d_c . That said, swarm seismicity does not always occur, and Supplementary Fig. 2a shows an example where fluid-driven aseismic slip propagates through the entire seismogenic zone. It seems there are more complex controls on this behavior, which we don't yet fully understand. We suspect that slight differences in initial shear stress might lead to these different behaviors, and are carrying out a controlled set of simulations to test this hypothesis. But this will be reported in another paper.

3) The first paragraph on p. 5 is somewhat confusing. Strictly speaking, from a hydrological point of view, "flow" occurs in response to pressure gradients. That redistribution of fluid mass does, ultimately, redistribute pressure. But I don't think it's totally accurate to say that the "upward flow...leads to overpressure". It would be better to say that the prescribed upward flux at the model base leads to overpressure.

Author response: Yes, we weren't careful with the wording in the original manuscript. Now we write "Note that $q = 0$ for the hydrostatic condition $p = \rho g z$, whereas fluid overpressure leads to upward flow ($q > 0$): $p = (\rho g + \eta q/k)z$ for constant q and k ."

4) Top of page 12 – these two paragraphs highlight the main and novel contribution of the work in my estimation. I think this needs to be brought out more strongly and emphasized both in the text as a primary result, and in the figures (see next suggestion). To some extent this also ties to

previous comment #2, which speaks to how robust this result is, and/or how strongly this behavior hinges on parameter choices.

Author response: The revised manuscript is extensively rewritten to highlight the two phenomena that appear in our simulations: fluid-driven aseismic slip fronts and swarm seismicity, both of which occur as overpressure pulses ascend through the crust. We feel that the “story” is less about precursory aseismic slip prior to nucleation of large events (which seems to be how you read the original manuscript) than about how these two phenomena occur due to coupling between fluids and frictional slip. In particular, fluid-driven aseismic slip does not always lead to large earthquake nucleation; as Fig. 2a shows, sometimes these aseismic slip fronts arrest (e.g., the shallower one around time step 2000) and other times the seismic event that is triggered is small. We feel that these new results, with smaller dc and nucleation length, are more realistic (and robust with respect to parameter choices) than our original simulations in which the large dc and nucleation length led to periodic earthquakes that ruptured the entire seismogenic zone.

5) Related to #4, I found that the graphics do show the key result, but the fundamental new insight might be emphasized with an additional or alternate graphic so it isn't lost in the details. Specifically, why not distill the result with a plot showing the depth of locking as $f(\text{time})$ for a few different parameter choices? This would illustrate both the sensitivity noted above, and bring forward the important idea that the upward-percolating fluids “erode” the locked zone from below. Likewise, I think a plot of the amount of creep (or slip rate at a couple of distinct depths) as $f(t)$ for a couple of different depths would be useful. The bottom line is that there are some important and really interesting outcomes here, and I think they are discernible but perhaps a bit lost in the details of figure 2, because there is a lot shown there (and because color contour plots are not as effective in conveying quantitative information as line plots).

Author response: Thanks for the suggestions. We have added the plot of locking depth vs. time for different healing times T (Fig. 6), highlighting the sensitivity of migration speeds to T . Furthermore, we also provide a new plot showing time histories of slip velocity, permeability, etc., at several depths (Supplementary Fig. 1). Finally, we also provide Supplementary Movie 1, which very clearly reveals the coupling between the different fields in a format that is probably easier to understand than the contours in Figs. 2 and 3. We feel that with these new additions, and accompanying changes to the text, the reader will find it much easier to understand the main results of our study.

6) I almost hate to say it, but a cartoon emphasizing the observations discussed in the last section of the paper (bottom of p. 12 through p. 13) might be helpful. The ideas and observations introduced in this section varied – ranging from gradual unlocking (this is explicitly reproduced in the models), to SSE (this is not...so a better discussion of how this might be linked to the models is needed, and particularly how the much faster rates of “unzipping”, on the order of km/d rather than tens of m/yr, would – or could – be explained by the models), to transient creep events on the SAF (which is linked only speculatively to the modeling results).

Broadly speaking, I think this section of the paper is a key element in engaging a broader audience and demonstrating the role that fluid migration may play in driving interseismic /precursory fault slip – so should be strengthened through more specificity and some direct comparisons of rates, timescales, etc...

Author response: We rewrote the Discussion (and other parts of the manuscript) to highlight the new results. We decided against adding a cartoon because we feel that the new figures and movie and text are effective at explaining the new results. We left a (modified) version of the paragraph about slow slip events (see response to another comment above) but decided to remove the paragraph about transient creep events on the San Andreas. This is because we have not investigated conditions that are relevant to the creeping section of the San Andreas (i.e., extensive velocity-strengthening regions), and while some authors have argued for a connection between fluids and transient creep on the San Andreas, our current models have little to say about this. We also added a new paragraph discussing swarm seismicity, but again it is difficult to directly compare our model to observations. We view our simulations as highlighting, in a generic context, how the included processes and feedbacks lead to certain phenomena (fluid-driven aseismic slip and swarm seismicity), rather than being realistic models of the San Andreas or Alpine Faults or some specific earthquake swarm sequence. We hope that the ideas showcased in our manuscript will prompt other modeling studies of specific events and faults.

Minor comments:

a) Abstract, lines 4-6. Seems more appropriate to say “...quantify the effects of fault valving through 2D...friction, in the face of upward flux of fluids from the mantle upward Darcy flow along a permeable fault zone, and permeability evolution.” [see also comment #3 above]

Author response: No change made. We prefer to leave the fluid source unspecified, and due to abstract length limitations we don't see the benefit of adding “the effects of” prior to “fault valving.”

b) Abstract, line 9: replace “shearing of the fault zone” with “slip”?

Author response: We now just say “earthquakes enhance permeability.” (It was necessary to shorten the abstract considerably to meet the Nature Communications requirement.)

c) P. 2, line 1: end of abstract: isn't “fault unlocking in the late interseismic period” basically “precursory slow [or aseismic] slip”? It seems clearer and more emphatic to say so if that's the case.

Author response: We revised the abstract to highlight both fluid-driven aseismic slip and swarm seismicity. We do not feel that the “story” is about precursory slip, given that the aseismic slip fronts do not always trigger a large earthquake.

d) P. 2: the rock mechanics community usually uses μ for friction coefficient. That may be problematic because the same symbol is also used for fluid viscosity, but worth thinking about the audience here.

Author response: We switched to using eta for viscosity.

e) P. 2 end of first paragraph of main text: dilatancy hardening has also been explored – not just thermal pressurization.

Author response: We added many new references to the introduction about previous work on fluids and faulting, including the Segall and Rice (1995) study of dilatancy hardening.

f) End of that same paragraph and elsewhere – it's awkward and not very useful to use "etc.". Suggest deleting, or replacing with specific list of items or at least a clearer articulation of the kinds of things that are tuned. Might be worth citing Liu & Rice for this statement also.

Author response: We removed most instances of "etc." and replaced them with more specific lists. We added Liu and Rice (2005) and other examples of studies where effective normal stress was used as a tuning parameter.

g) P. 2, 4th line from bottom: "Flow along faults creates pressure gradients" isn't quite accurate – see comment #3 above. Flow occurs in response to pressure gradients and ultimately serves to dissipate them.

Author response: See response to comment above. We modified the wording.

h) Second to last line: delete the word "perhaps"

Author response: Corrected.

i) Last line: replace "the high rates" simply with "to"

Author response: Corrected.

j) P. 3, first line: overpressure doesn't result from compaction and lithification. It's the other way around – those changes in the rock or sediment occur in response to flow of fluids and escape of overpressure. The water is pressurized by mechanical loading (burial + tectonic stress), and then flows away (i.e. dewatering).

Author response: We now say "overpressure from burial of sediments."

k) Last paragraph of intro: Warren Smith (2019) just published a paper on valving related to SSE in New Zealand that should be cited here.

Author response: Added.

l) P. 4 first full paragraph: seems important to either justify the argument that country rock is “impermeable” or discuss briefly how fluid leakage into the wall rock might impact the results.

Author response: See response to a similar reviewer comment above regarding fault-normal pressure gradients and flow.

m) P. 4 bottom: see “f” above, use of “etc.”

Author response: Corrected.

n) P. 5 top: the fluxes observed on continental plate boundary faults are highly focused at springs and other discharge points rather than uniformly distributed along the fault length. This would imply that at depth, the flux (per m of fault length) is smaller, and this then is “collected” somehow along the path from deep crust to surface.

Author response: The fluid flux varies quite a lot among different studies in the literature, and our objective here was to choose a somewhat representative value while focusing on generic, qualitative phenomena that arise from the coupling between fluids and frictional slip. We look forward to continuing this research effort by studying different fluxes, extending the model to account for porous flow in the bulk, and moving from 2D to 3D. The latter two model extensions would allow us to study how various nonlinearities and feedbacks of the system lead to flow channelization.

o) P. 5, below eq (3): 10-19 could be considered a pretty high number for k of lower crustal rock, even faults. When effective stress is high (i.e. the “minimum bound”), a number 10-100 times lower would be reasonable (see Lockner and his group’s work on Nojiema and SAF rock). The numbers from Manning & Ingebritsen (1999) are long-term averages and likely represent mainly flux during phases of elevated pore pressure driven by metamorphic fluid release.

p) P. 5 after equation 4: likewise, the choice of upper bound on k is probably low. Observations of post-seismic fluid discharge, along with modeling studies constrained by data indicating thermal or geochemical advection, suggest k values that are ~10-1000 times higher. It would be interesting to see if this would drive more rapid migration of aseismic slip fronts (see point #6 above – particularly as related to the question of explaining SSE which “unzip” much faster than a few tens of m/yr).

Author response for (o) and (p): We agree that the range of permeability can be larger than the range used in our study. Even with the more conservative permeability range we adopted, observed cyclic overpressure build-up and release, fluid-driven aseismic slip, and even swarm seismicity. We do suspect that larger changes in permeability, as well as higher fluid fluxes, will increase the propagation rates of aseismic slip fronts, possibly into the range of slow slip events. But we defer this for future studies.

q) P. 6 and more generally: how is normal stress defined? Is it simply a Poisson effect, or is tectonic loading included?

Author response: We follow the same setting as Allison and Dunham (2018): “The normal stress is determined for a optimally oriented strike-slip fault by assuming that the vertical total stress is equal to lithostatic pressure and the ratio of fault shear stress to effective normal stress is approximately equal to the reference coefficient of friction $f_0=0.6$.”

r) P. 8, statement “...leads to an earthquake.” should be referenced.

Author response: We have extensively rewritten this section of the manuscript, so this comment isn't really valid anymore.

s) P. 8: would be useful to make clear in the main text whether k increases only in the part of the fault that slips, or if there is some damage assumed in a halo or region that extends some distance from the actively slipping patches ends.

Author response: Based on eq. (4), permeability increases only in the part of the fault that slips. In reality, the region of elevated permeability would extend some distance ahead (and around) the slip front, if permeability enhancement comes from cracking or yielding processes in the fault damage zone caused by the stress concentration at the rupture front. We thought about using nonlocal permeability enhancement, but it seems overly complicated. We prefer to do this problem more rigorously, in the future, by explicitly resolving the fault damage zone and solving the fluid transport equations in 2D (or 3D) so we can account for the finite width of the damage zone and other details of the evolving permeability structure of real faults and fault-normal flow.

t) P. 12: It's a bit nuanced, and maybe doesn't matter hugely, but generally speaking, if the fault is healing hydrologically, it implies a reduction in porosity (cracks or intergranular). This same process should somehow be connected to frictional healing through the parameter b . By that logic, one might expect the value of RSF parameter b (high values indicating greater healing) to map at least qualitatively to the rate of permeability increase. Likewise where b is small (i.e. greater depths in the model space as shown in Fig 1), contact area and porosity don't evolve or heal much, so this ought to map to decreased rates of change of k .

Author response: This is possibly correct, and some recent experiments (Im et al., 2018) demonstrate that permeability healing and fractional healing might arise from a common mechanism at the asperity contact scale. In addition, it is undoubtedly the case that healing and sealing processes are temperature and hence depth dependent, either for the reason you pointed out or for other reasons. However, for simplicity we decided to adopt a single healing time across all depths. Future work will consider depth-dependent healing/sealing but that will require identifying the relevant mechanical and/or chemical processes responsible for porosity and permeability changes and also the relation between porosity and permeability.

u) P. 13, middle paragraph, last sentence: this would also occur if there are lateral changes in (b-a) which may be as, or more, likely than variations in fluid production rate.

Author response: Good point, we now write “lateral variations in frictional properties, fluid production rate, ...”

v) Same paragraph, last sentence: doesn't this also fundamentally require a transition from negative to positive (b-a)? I think that's what the models are showing – that aseismic failure takes place in regions that were previously locked, but were never going to nucleate instability anyway because they are rate weakening?

Author response: It seems that fluid-driven aseismic slip can occur in both velocity-weakening and velocity-strengthening regions. However, given that the frictional transition takes place at 17 km depth in our model, we don't really have much insight into how this phenomenon occurs in velocity-strengthening parts of the fault.

References:

- Allison, K. L., & Dunham, E. M. (2018). Earthquake cycle simulations with rate-and-state friction and power-law viscoelasticity. *Tectonophysics*, 733, 232-256.
- Bense, V. F., Gleeson, T., Loveless, S. E., Bour, O., & Scibek, J. (2013). Fault zone hydrogeology. *Earth-Science Reviews*, 127, 171-192.
- Caine, J. S., Evans, J. P., & Forster, C. B. (1996). Fault zone architecture and permeability structure. *Geology*, 24(11), 1025-1028.
- Erickson, B. A., Jiang, J., Barall, M., Lapusta, N., Dunham, E. M., Harris, R., ... & Cattania, C. (2020). The Community Code Verification Exercise for Simulating Sequences of Earthquakes and Aseismic Slip (SEAS). *Seismological Research Letters*, 91(2A), 874-890.
- Im, K., Elsworth, D., & Fang, Y. (2018). The influence of preslip sealing on the permeability evolution of fractures and faults. *Geophysical Research Letters*, 45(1), 166-175.
- Lapusta, N., Rice, J. R., Ben-Zion, Y., & Zheng, G. (2000). Elastodynamic analysis for slow tectonic loading with spontaneous rupture episodes on faults with rate-and state-dependent friction. *Journal of Geophysical Research: Solid Earth*, 105(B10), 23765-23789.

Liu, Y., & Rice, J. R. (2005). Aseismic slip transients emerge spontaneously in three-dimensional rate and state modeling of subduction earthquake sequences. *Journal of Geophysical Research: Solid Earth*, 110(B8).

Rice, J. R. (1992). Fault stress states, pore pressure distributions, and the weakness of the San Andreas fault. In *International geophysics* (Vol. 51, pp. 475-503). Academic Press.

REVIEWERS' COMMENTS:

Reviewer #1 (Remarks to the Author):

The authors have addressed all of my comments in redrafting their submission. I did, however, again go through the revised draft once more and found a few issues that I mention below. I strongly feel that addressing these comments need not constitute a revision, and hence, would not want to look again at the authors' responses. I trust the authors to address these issues satisfiably.

- 1) Page 11, last paragraph: I felt that the inline equation has a typo, it should be $q_{max} \approx \frac{k_{max}}{\eta} \left(\frac{\partial p}{\partial z} - \rho g \right)$. You have σ inside the bracket and the derivative is total. Am I missing something? Additionally, I felt that the comment about integrating eq. (2) leading to the expression $\frac{dp}{dt} \approx -\frac{q_{max}}{n\beta H}$ is a bit problematic unless you assume that vertical variations in the depressurization rates are small over the length scale H . It is difficult to gauge this from the plots, but from Figure 3b and supplementary figure 4b, the spacing between the grey lines seems to vary as a function of depth. Is it better to just call this a dimensional estimate of the depressurization rate?

And finally, in estimating the characteristic pressure drop, why does it make sense to assume the sealing time scale T as the appropriate time scale for depressurization? Is depressurization not driven mostly by slip induced permeability enhancement – in that case, does it make sense to assume L/V_p as the upper bound of the time scale to calculate the pressure drop?

- 2) Below equation 1, the authors point out that β is the sum of fluid and pore compressibility. Usually, the volume water retention in the medium is modeled as a storage coefficient of the form $S_s = \alpha + n\beta$ where α is a matrix compressibility (for example, see Dingman, 2015, Physical Hydrology). I am slightly confused as why the matrix compressibility is completely neglected here – the usual approximation is to assume the fluid as relatively incompressible ($\beta \ll \alpha$), here the authors seem to have assumed the opposite, i.e. the matrix is relatively incompressible. Just wanted to make sure that this is not a typo.

Additionally, written as it is, β also has a contribution from pore compressibility. I am unsure as to what is conveyed by pore compressibility. It would be good to see at least one standard reference for this unusual mathematical formulation and a brief explanation for the physical implications of this formulation.

- 3) The authors seem to indicate that by using a smaller d_c they were able to observe the swarm like earthquake phenomena. If the explanation of the swarm seismicity in the smaller d_c simulations is indeed a smaller nucleation length, would the authors not have observed the same phenomena with the larger d_c as long as the ratio of d_c to $\sigma - p$ was the same as the present simulations? In fact, as the authors know, the nucleation length scale (for the Aging law) is determined by the ratio of the effective normal stress to d_c and also by the ratio of a/b and b . Does it make sense to add a couple sentences in the conclusions to this effect – that a smaller d_c might not be the sole requirement for the emergence of swarm activity. However, I do acknowledge the fact that the smaller d_c is more consistent with laboratory experiments.

With these issues addressed, I think the paper is fit to be published. The natural emergence of earthquake swarms in this model particularly adds a significant chunk to the paper and is definitely worthy of publication in Nature Communications.

Pathikrit Bhattacharya

Reviewer #2 (Remarks to the Author):

The revision has accounted for the bulk of my comments, by explaining the parameter choices and exploring a wider range of parameters, and by modifying some of the discussion. The study is an insightful and valuable contribution that uses novel numerical modeling with combined evolution of fault slip and permeability to demonstrate the feasibility of the fault-valving hypothesis and explore its consequences in terms of slip behavior. The hypothesis provides a potential explanation to significant, near-lithostatic, fluid overpressure hypothesized, based on a range of observations, at the bottom of seismogenic portions of subduction zones and at the roots of some strike-slip faults as referenced in the manuscript. The study highlights a range of interesting fault behaviors that result and compares them to observations of aseismic slip and earthquake swarms. The manuscript is well written and will be thought provoking for a wide audience interested in earthquake source processes. I recommend that it be published in its present form.

We thank the reviewers for the valuable feedback to help improve our manuscript. Below is our point-by-point response:

Reviewer #1:

The authors have addressed all of my comments in redrafting their submission. I did, however, again go through the revised draft once more and found a few issues that I mention below. I strongly feel that addressing these comments need not constitute a revision, and hence, would not want to look again at the authors' responses. I trust the authors to address these issues satisfiably.

Author response: We thank the reviewer for the time and effort invested in the review of our manuscript.

1). Page 11, last paragraph: I felt that the inline equation has a typo, it should be $q_{max} \approx \frac{k_{max}}{\eta} (\frac{\partial p}{\partial z} - \rho g)$. You have σ inside the bracket and the derivative is total. Am I missing something? Additionally, I felt that the comment about integrating eq. (2) leading to the expression $\frac{dp}{dt} \approx -\frac{q_{max}}{\eta\beta H}$ is a bit problematic unless you assume that vertical variations in the depressurization rates are small over the length scale H . It is difficult to gauge this from the plots, but from Figure 3b and supplementary figure 4b, the spacing between the grey lines seems to vary as a function of depth. Is it better to just call this a dimensional estimate of the depressurization rate?

Author response:

- (a) $d\sigma/dz$ is correct as written. You are correct that the maximum permeability k_{max} should be used in Darcy's law, but we also need to approximate the pressure gradient. Here we follow the logic explained in the previous sentence, "the pressure gradient is bounded approximately by the fault normal stress gradient", i.e., $\partial p/\partial z \approx d\sigma/dz$. This bounding condition comes from the fact that effective normal stress must be nonnegative (and hydraulic fracturing and loss of overpressure would probably occur prior to the effective normal stress reaching zero). A total derivative on normal stress is appropriate because (in this model) it depends only on depth z .
- (b) The depressurization rate is estimated by integrating the mass balance equation (1) over the seismogenic zone. You are correct that the estimate is most justified if the depressurization rate is relatively uniform over the seismogenic depth, making this an approximation (which is partially why we used the approximation symbol rather than equality). We added the parenthetical remark "(and neglecting spatial variations in depressurization rate)" to clarify this approximation. With the simplifying approximation now stated clearly, we do not feel it necessary to change the notation or to otherwise indicate that this is a dimensional estimate (as it is better than that, but subject to the approximation).

And finally, in estimating the characteristic pressure drop, why does it make sense to assume the sealing time scale T as the appropriate time scale for depressurization? Is depressurization

not driven mostly by slip induced permeability enhancement – in that case, does it make sense to assume L/V_p as the upper bound of the time scale to calculate the pressure drop?

Author response: The depressurization process starts in the post-rupture enhanced permeability state and ends when the permeability has dropped by a few orders of magnitude. That permeability reduction is primarily controlled by healing/sealing (second term on the right-hand side of the permeability evolution equation (4)), rather than the slip-dependent permeability enhancement mechanism (first term in the permeability evolution equation). Healing/sealing occurs on time scale T , thus making this the relevant time scale for depressurization duration. As evidence of this, compare the four simulations in the manuscript, which have four different time scales T but the same L/V_p . You can see that these simulations all have very different depressurization time scales that are controlled by the healing/sealing time T .

2) Below equation 1, the authors point out that β is the sum of fluid and pore compressibility. Usually, the volume water retention in the medium is modeled as a storage coefficient of the form $S_s = \alpha + n\beta$ where α is a matrix compressibility (for example, see Dingman, 2015, Physical Hydrology). I am slightly confused as why the matrix compressibility is completely neglected here – the usual approximation is to assume the fluid as relatively incompressible ($\beta \ll \alpha$), here the authors seem to have assumed the opposite, i.e. the matrix is relatively incompressible. Just wanted to make sure that this is not a typo.

Additionally, written as it is, β also has a contribution from pore compressibility. I am unsure as to what is conveyed by pore compressibility. It would be good to see at least one standard reference for this unusual mathematical formulation and a brief explanation for the physical implications of this formulation.

Author response: Thanks for suggesting that we clarify our assumptions in the porous flow model. We are not neglecting matrix compressibility here, and we believe that the term “pore compressibility” is standard in the poroelasticity literature. We now provide references to Segall and Rice (1995) and Rice (2006) that utilize a similar derivation and notation. You can also find a definition of pore compressibility in “The Rock Physics Handbook” by Mavko et al. (2020) on p. 180. When the fluid mass content, per unit reference volume of porous material, is written as the product of fluid density and pore volume fraction ($m = \rho n$), then changes in fluid mass content arise from changes in fluid density and pore volume fraction ($dm = nd\rho + \rho dn$). The relative (elastic) change in pore volume fraction arising from a change in pore pressure is defined as the pore compressibility ($\beta_n = n^{-1}dn/dp$). (There are a variety of pore compressibilities that can be defined depending on whether confining stress or some combination of total stress and strain components is held fixed when pore pressure is changed; see discussion in Appendix A of Rice (2006).) It is also possible to rewrite the pore compressibility in terms of matrix compressibility (defined as the relative change in total volume of porous material under fixed pore pressure or drained conditions) and other parameters like pore volume fraction (as in the expression you cited); however, we feel it is unnecessary to utilize such relationships in this study, and we prefer to utilize the pore compressibility itself without getting into details of what controls it.

3) The authors seem to indicate that by using a smaller d_c they were able to observe the swarm like earthquake phenomena. If the explanation of the swarm seismicity in the smaller d_c simulations is indeed a smaller nucleation length, would the authors not have observed the same phenomena with the larger d_c as long as the ratio of d_c to $\sigma - p$ was the same as the present simulations? In fact, as the authors know, the nucleation length scale (for the Aging law) is determined by the ratio of the effective normal stress to d_c and also by the ratio of a/b and b . Does it make sense to add a couple sentences in the conclusions to this effect – that a smaller d_c might not be the sole requirement for the emergence of swarm activity. However, I do acknowledge the fact that the smaller d_c is more consistent with laboratory experiments.

Author response: We totally agree that a smaller d_c is not the sole requirement for the emergence of swarm activity; it is really that nucleation length is sufficiently small. To clarify, rewrote this portion of the sentence and now provide a reference for readers to learn more about controls on nucleation length: "...swarm seismicity requires rate-weakening friction and sufficiently small nucleation length (e.g., from sufficiently small state evolution distance and/or high effective stress; for specific parameters influencing nucleation length, see Rubin (2008))..."

With these issues addressed, I think the paper is fit to be published. The natural emergence of earthquake swarms in this model particularly adds a significant chunk to the paper and is definitely worthy of publication in Nature Communications.

Reviewer #2:

The revision has accounted for the bulk of my comments, by explaining the parameter choices and exploring a wider range of parameters, and by modifying some of the discussion. The study is an insightful and valuable contribution that uses novel numerical modeling with combined evolution of fault slip and permeability to demonstrate the feasibility of the fault-valving hypothesis and explore its consequences in terms of slip behavior. The hypothesis provides a potential explanation to significant, near-lithostatic, fluid overpressure hypothesized, based on a range of observations, at the bottom of seismogenic portions of subduction zones and at the roots of some strike-slip faults as referenced in the manuscript. The study highlights a range of interesting fault behaviors that result and compares them to observations of aseismic slip and earthquake swarms. The manuscript is well written and will be thought provoking for a wide audience interested in earthquake source processes. I recommend that it be published in its present form.

Author response: We thank the reviewer for recognizing the broad applicability of this modeling framework and supporting its publication.